# Suppression of gut colonization by multidrug-resistant *Escherichia coli* clinical isolates through cooperative niche exclusion

Marie Wende[1,2], Lisa Osbelt [1,3,11], Lea Eisenhard[1,11], Till Robin Lesker[1], Bamu F. Damaris[4], Uthayakumar Mutukumarasamy[1], Agata Bielecka[1], Éva d. H. Almási [1], Katrin Anja Winter [1], Jennifer Schauer[5], Niels Pfennigwerth[5], Sören Gatermann [5], Katharina Schaufler [6,7], Dirk Schlüter [8,9], Marco Galardini [4,8] & Till Strowig [1,3,8,10] ✉

Human gut colonization by multi-drug resistant Enterobacterales (MDR-E) poses a risk for subsequent infections. Because of the collateral damage antibiotics cause to the microbiota, microbiome-based interventions aimed at promoting decolonization have garnered interest. In this study, we evaluate the strain-specific potential of 430 commensal *Escherichia coli* isolates to inhibit the growth of an MDR *E. coli* strain. Comparative analyses using in vitro, ex vivo, and mouse models reveal that only a subset of commensal strains can facilitate gut decolonization. Bioinformatic and experimental analyses of the antagonism among representative strains demonstrate that both direct and indirect carbohydrate competition contribute to niche exclusion between *E. coli* strains. Finally, the combination of a protective *E. coli* strain with a *Klebsiella oxytoca* strain enhances the inhibitory potential against metabolically diverse MDR *E. coli* strains and additional MDR-E species, highlighting that rationally designed metabolically complementary approaches can contribute to developing next-generation probiotics with broad-spectrum activity.

The emergence and spread of antibiotic resistance is a complex and multifaceted medical and societal challenge with immediate consequences on patient mortality, length of hospital stay, and respective healthcare costs[1]. Statistical models have estimated that 4.95 million deaths can be associated with bacterial antimicrobial resistance (AMR) in 2019[2]. The pathogen with the highest estimated deaths associated with AMR is *Escherichia coli* (*E. coli*), with approximately 800,000 directly attributed global deaths in 2019[2]. The most critical AMR in Enterobacterales are extended-spectrum β-lactamases (ESBL) and carbapenem resistance[3–5], which frequently require treatment with "last-resort" antibiotics or are even untreatable[6]. One critical risk factor for infection with such bacteria is gut colonization, which serves as a

[1]Department of Microbial Immune Regulation, Helmholtz Center for Infection Research, Braunschweig, Germany. [2]ESF International Graduate School on Analysis, Imaging and Modelling of Neuronal and Inflammatory Processes, Otto-von-Guericke University, Magdeburg, Germany. [3]German Center for Infection Research (DZIF), Braunschweig, Germany. [4]Institute for Molecular Bacteriology, TWINCORE Centre for Experimental and Clinical Infection Research, a joint venture between the Hannover Medical School (MHH) and the Helmholtz Centre for Infection Research (HZI), Hannover, Germany. [5]National Reference Centre for Multidrug-resistant Gram-negative Bacteria, Department of Medical Microbiology, Ruhr-University Bochum, Bochum, Germany. [6]Helmholtz Institute for One Health, Helmholtz Center for Infection Research HZI, Department of Epidemiology and Ecology of Antimicrobial Resistance, Greifswald, Germany. [7]University Medicine Greifswald, Greifswald, Germany. [8]Cluster of Excellence RESIST (EXC 2155), Hannover Medical School, Hannover, Germany. [9]Institute of Medical Microbiology and Hospital Epidemiology, Hannover Medical School, Hannover, Germany. [10]Center for Individualized Infection Medicine (CiiM), a joint venture between the Helmholtz-Center for Infection Research (HZI) and the Hannover Medical School (MHH), Hannover, Germany. [11]These authors contributed equally: Lisa Osbelt, Lea Eisenhard. ✉e-mail: till.strowig@helmholtz-hzi.de

reservoir for spreading and translocating into the bloodstream[7]. Concomitantly, fecal carriage of ESBL-producing *E. coli* has increased substantially in the global community among healthy individuals, rising from 2.6% (2003–2005) to 21.1% (2015–2018) in some areas of the world[8]. In hospitalized individuals, colonization rates are even higher (25.7%, 2016–2020)[9] with ESBL-producing *E. coli* being the most prominent ESBL-producing species of Enterobacterales[10]. The genetically highly diverse species of *E. coli* consists of 8 phylogroups[11,12] and more than 13,000 sequence types (STs)[13,14], with a subset of STs being predominantly associated with AMR and nosocomial infections. The most prevalent ST causing the majority of infections of MDR *E. coli* is ST131[15,16], followed by others such as ST38, ST410[17,18], ST167, and ST617[19]. Thus, these STs are of utmost importance and should be prioritized for targeted eradication.

Colonization resistance describes the ability of the healthy microbiota to prevent the expansion of potential pathogens by direct microorganism-mediated inhibition or indirectly by enhancing immunity by stimulation of the mucosal immune system[20]. Under homeostatic conditions Enterobacterales are a minor part of the adult colonic microbiota, typically representing <1% of the biomass; however, disruption of the microbiota by antibiotics and non-antibiotic drugs leads to a loss of colonization resistance[21,22] and allows drug-resistant Enterobacterales to expand to higher densities in the gastrointestinal (GI) tract, significantly elevating the risk of infections[23,24]. Direct mechanisms of colonization resistance involve, for instance, the production of specific inhibitory metabolites like bacteriocins[25,26] and short-chain fatty acids[24,27], or the metabolism of bile salts[28]. Another important mechanism is metabolic competition, i.e., the so-called niche-exclusion principle, through which one species depletes specific critical nutrients like amino acids[29,30] or carbon sources[31–35] that another species requires to establish itself within the community. Niche exclusion by carbohydrate competition is commonly described between related species or strains among Enterobacterales. For example, specific strains of *Salmonella* Typhimurium (*S.* Typhimurium) and *E. coli* or *Klebsiella oxytoca* (*K. oxytoca*) compete for the sugar alcohol dulcitol[32,34,36]. *K. oxytoca* and *Klebsiella pneumoniae* compete for beta-glucosides[31], and *K. oxytoca* species complex strains have been shown to compete with *E. coli* and *S.* Typhimurium[35–37]. Therefore, colonization resistance against major pathogens, especially after major disturbances such as broad-spectrum antibiotics, requires an efficient limitation of available nutrients and niches, ideally by species or strains with diverse metabolic capacities[34,38]. This principle of nutrient deprivation is widely acknowledged as a key aspect of a robust microbiota. However, the characteristics of individual strains and species within intricate communities remain largely unexplored despite their essentiality in niche exclusion after antibiotic-caused disturbance of the microbiota. For instance, we and others recently demonstrated a dependency on the microbial community to restore full colonization resistance, compared to only reduced colonization levels[31,32,39]. Therefore, defined bacterial consortia are being investigated regarding their ability to enable colonization resistance. For example, colonization resistance against vancomycin-resistant *Enterococcus* (VRE) can be enhanced by *Blautia producta* and *Clostridium boltae*[40], and microbial consortia can protect against *Clostridioides difficile* or *S.* Typhimurium infections[39,41,42]. The strain *E. coli* Nissle 1917 (EcN), which has been sold as a dietary supplement for decades, exerts in animal models protective effects against pathogenic Enterobacterales like *S.* Typhimurium[26] or *E. coli*[43,44] via multiple mechanisms including microcin and siderophore production. However, its effectiveness as a single agent in decolonizing MDR *E. coli* is limited[45], and the production of the genotoxin colibactin by this strain is even associated with an elevated risk for colorectal cancer in animal models[46,47]. Despite these limitations, there has been a growing interest in commensal enterobacterial strains, specifically *E. coli* strains, in strategically designed probiotic cocktails, as Enterobacterales fulfill

critical roles in niche restriction[31,32,34,36,39]. However, the utility of commensal *E. coli* strains for promoting decolonization of MDR-E has not been systematically investigated.

To identify commensal strains with competitive characteristics against critical MDR *E. coli* STs, we generated a strain collection consisting of 430 *E. coli* strains isolated from fecal samples of human donors. Each member of the strain collection was screened regarding its genomic features. We demonstrated that candidate probiotic strains enable the decolonization of MDR *E. coli* in mice, depending on the microbial context. Furthermore, analysis of the carbohydrate utilization patterns and competition for specific carbohydrates revealed that candidate probiotic strains outcompete MDR *E. coli* via metabolic niche exclusion. Finally, the rational combination of protective *E. coli* with a metabolically complementary *K. oxytoca* strain could expand the target spectrum beyond MDR *E. coli*. These findings underline the potential of specific microbial combinations as probiotics for the decolonization of MDR-E.

## Results

### Commensal *E. coli* strains with inhibitory effects against an MDR *E. coli* strain are enriched in phylogroups B1 and D

To investigate the ability of intestinal *E. coli* strains originating from the general community to displace clinical isolates of MDR *E. coli*, we generated a comprehensive strain collection of Enterobacterales. In brief, fecal samples of non-hospitalized human donors (*n* = 630) from children and adults were selectively plated, and strains were identified using colony morphology (one colony per morphology for each donor was selected). Their taxonomy was assigned using 16S rRNA gene sequencing. *E. coli* was the most prevalent species, followed by *Klebsiella* spp., other *Escherichia* spp., *Enterobacter* spp., and *Citrobacter* spp. (Fig. S1A, Table S1). In the following paragraphs, these isolates are referred to as commensal since they are derived from the general (non-hospitalized) community, even though it can not be excluded that some of these strains have pathogenic or MDR characteristics. In total, 439 strains of *E. coli* were isolated (total number of strains *n* = 678) from different donors, and only one strain of *E. coli* per donor was included in the strain collection (Fig. S1B). Subsequently, *E. coli* strains (*n* = 430) were quantitatively evaluated for their competitive effects using an ex vivo screening assay. In brief, we co-cultivated an MDR *E. coli* strain in the presence or absence of individual strains from the strain collection in a 1:10 (MDR:commensal, Fig. S1C) ratio under anaerobic conditions, using an Anaeropack, in a media based on the cecum content of germ-free (GF) mice imitating the nutritional landscape of the gut environment without the interference of other bacteria[31,36] (Fig. 1A). We selected an MDR *E. coli* strain (from hereon referred to as MDR1, Table S2) belonging to ST617 and phylogroup A for this assay, which was isolated from a rectal swab and encodes for the ESBL OXA-48 and NDM-1. As a reference strain, *E. coli* Nissle 1917 (EcN) was included in this screening. *E. coli* MDR1 growth in single and co-culture was quantified using selective plating and the fold change of growth (FCgrowth) in co-culture vs. single culture (control) was calculated.

Commensal strains showed a highly variable degree of competitive effects, ranging from a FCgrowth of 16.8 (growth promotion) to 0.004 (growth inhibition) (Fig. 1B, Supp. Data 1). We categorized the strains into competitive and non-competitive strains, *i.e.*, the 10% of strains (*n* = 43) mediating the strongest inhibition of MDR1 growth were defined as competitive strains (mean FCgrowth = 0.0276), and all other remaining strains were grouped as non-competitive strains (mean FCgrowth = 0.6699), although they display a large variance of competitive effects. No obvious differences in competitiveness could be observed in *E. coli* strains originating from different cohorts or age groups (Fig. S1D). Notably, the probiotic EcN did not belong to the group of competitive isolates with a FCgrowth of 0.25.

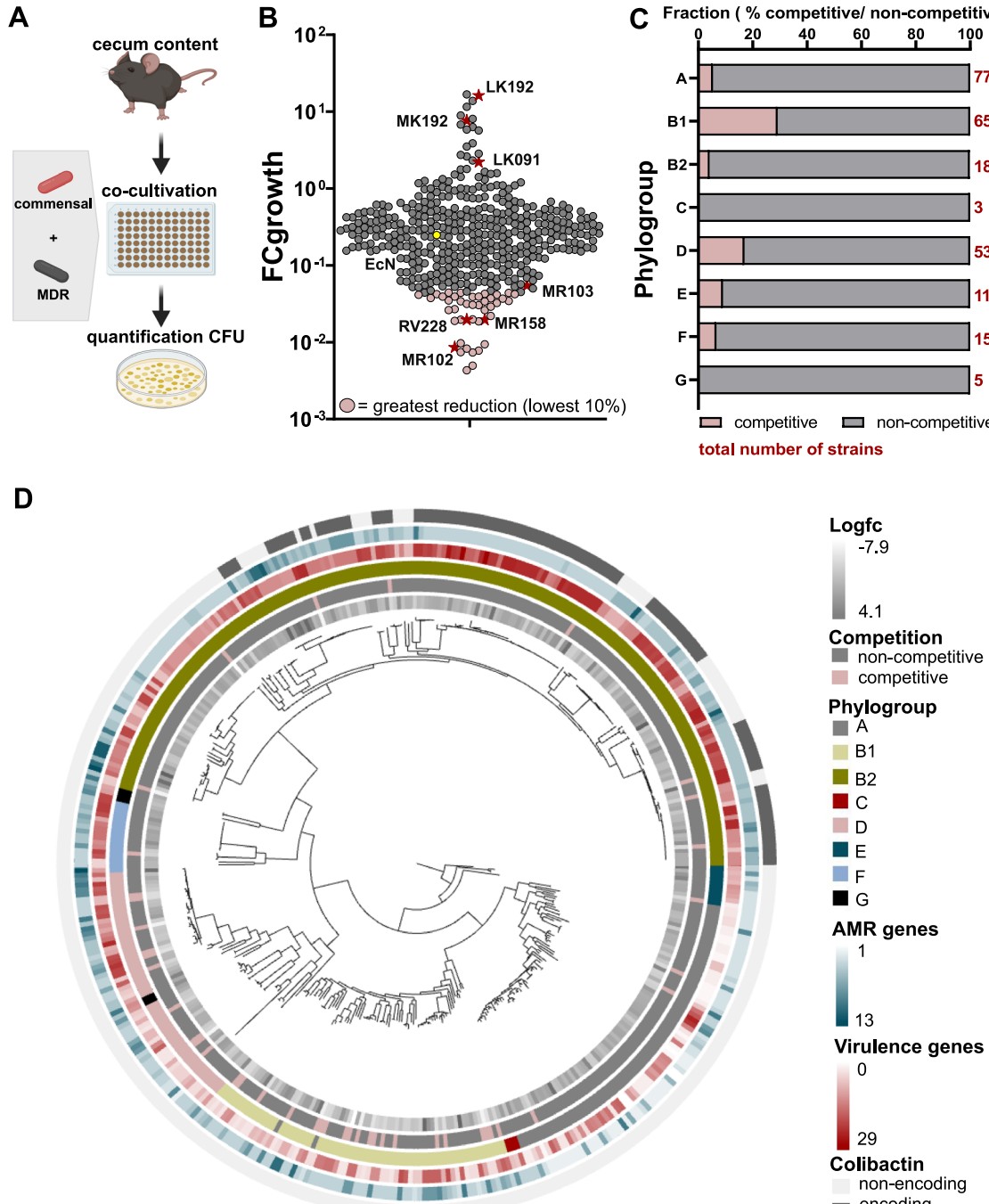

**Fig. 1 | Commensal *E. coli* strains show different competitive effects.**
**A** Schematic overview of experimental workflow. Commensal and MDR *E. coli* strains are spiked into isolated cecal contents of GF animals in a 10:1 ratio. After 24 h of anaerobic co-cultivation, CFUs of MDR *E. coli* were quantified by plating on selective agar plates. Created with Biorender. **B** Fold change of co-cultures to control of all strains ($n = 430$). Strains marked in red were selected for further in vivo experiments. **C** Fraction of competitive and non-competitive isolates in each phylogroup. **D** The phylogenetic relationship of commensal *E. coli* isolates is visualized in a taxonomic tree. Different rings (from outer to inner ring) show the presence of colibactin, virulence genes, AMR genes, phylogroup, competitive phenotype, and log-fold reduction.

Next, to analyze whether a correlation between phylogeny and the competitive phenotype could be observed, commensal strains were whole-genome sequenced followed by phylogenetic analysis. We observed at the level of coarse population structure that phylogroup B2 was the most prevalent, containing 42.8% ($n = 188$) of strains, followed by phylogroup A with 16.9% ($n = 74$). The phylogroups B1, D, F, E, G, and C were less frequent (Fig. 1C, D, Table S3). Competitive strains were enriched in phylogroups B1 and D with 29.2% ($n = 19/65$) and 17%

($n = 9/53$) of competitive strain per phylogroup, respectively, compared to 4.2% in phylogroup B2 ($n = 8/180$) (Chi-square test, **** = $p < 0.0001$) (Fig. 1C, D, S1E, Table S3). Within phylogroups, competitive and non-competitive strains were interspersed on the phylogenetic tree.

To evaluate potential probiotic bacteria, detailed information on genes encoding virulence factors and AMR is important. First, virulence-associated genes (VAG) and AMR genes (ARG) between

strains of different phylogroups were compared. Commensal *E. coli* strains from the phylogroups B2 and F tended to have a higher prevalence of VAGs, while only slight differences for ARGs (reduced levels in phylogroup A) could be observed (Fig. S1F, G). Nevertheless, a comparison of the commensal strains to a reference collection of clinical isolates[48] demonstrated that the number of both virulence (12.9 vs. 17.0) and AMR (5.1 vs 7.4) genes were on average significantly lower in the commensal vs. clinical isolates (virulence: cohen´s *d* 0.54, t-test *p*-value «10$^{-10}$; AMR: cohen´s *d* 0.88, t-test *p*-value «10$^{-10}$) (Fig. S1H, I, Supp. Data 2). Due to the enrichment of competitive strains in specific phylogroups, VAGs and ARGs were compared between competitive and non-competitive strains within phylogroups, with no statistically significant differences being detectable between competitive and non-competitive strains. Finally, colibactin has been identified as a virulence factor and genotoxin, promoting colorectal cancer, but has also been hypothesized to promote interspecies competition within the microbiota[46,49,50]. In phylogroup B2, 54.3% (102/188) of strains, including EcN, encoded the *pks* operon required for colibactin production. In line with other studies, no strain from the other phylotypes encoded the pks operon/island[51,52] (Supp. Data 1).

Together, these results demonstrate that commensal *E. coli* strains show different abilities to compete against MDR *E. coli* ex vivo, with phylogroup B1 and D being enriched in competitive isolates. The virulence gene profile analysis reveals a higher prevalence of virulence genes in phylogroup B2, but not between competitive and non-competitive strains.

### Strains with competitive effect ex vivo can decolonize MDR *E. coli* from the mouse gut

The in vitro assay has been previously utilized to identify strains of the *K. oxytoca* species complex (KoSC) with inhibitory potential against *S.* Typhimurium both in vitro and in vivo[36], but the utility for predicting the competitive potential of *E. coli* strains has not been tested. Therefore, a panel of commensal strains falling into the two categories was tested in a previously established prophylactic mouse gut decolonization model, which allows the testing of strains with naturally occurring ampicillin resistance[31]. In brief, SPF mice were pretreated with ampicillin (0.5 g/L in drinking water) for 4 days and then precolonized with the commensal strains (10$^8$ CFUs) via oral gavage, followed 4 days later by colonization with the *E. coli* MDR1 strain (10$^8$ CFUs) (Fig. 2A). Ampicillin administration was stopped 3 days later. We evaluated the following strains, all encoding endogenous ampicillin resistance cassettes (Supp. Data 2): *E. coli* MR102 (phylogroup D), MR158 (B1), and RV228 (D) as competitive strains as they belong to 10% of the strains with the greatest growth inhibition, and the strains MR103 (B1), LK91 (B2), MK192 (A), and LK192 (B2) as non-competitive strains. Fecal colonization levels of the commensal strains were monitored 4 days after precolonization and of the MDR1 strain longitudinally over six weeks (Fig. 2A). Of note, the commensal strains displayed colonization levels ranging from 10$^9$ to 10$^{11}$ CFU/g in the feces, with MK192 having the lowest precolonization levels and LK192 showing the highest variability between mice (Fig. S2A). Precolonization of mice with commensal *E. coli* strains led to a variable reduction of MDR1 colonization compared to the control group (Fig. 2B). A 100-fold reduction of colonization levels of MDR1 compared to control mice was observed at day 3 in mice precolonized with strains MR102, MR158, RV228, MR193, LK91, and MK192 and a 10-fold reduction in the group precolonized with LK192 (Fig. S2B). No spontaneous clearance (clearance = CFU/g below the detection limit in feces) of the MDR1 strain was observed in the control mice, demonstrating stable integration of this strain into the microbiota until the end of the experiment. In contrast, on day 9, clearance started in the groups precolonized with competitive isolates. For instance, 33.3% (5/15) of the mice precolonized with MR102 and 50 % (5/10) precolonized with RV228 cleared MDR1 from the feces (Fig. S2C). At the end of the

experiment (d42), clearance rates of more than 80% were observed in the groups precolonized with MR102 (13/15), MR158 (8/10), and RV228 (8/10). In contrast, in the groups precolonized with MR103 (4/9), LK91 (5/9), and MK192 (5/11), clearance rates of 40-60% and for LK192 a clearance rate of 10% (1/10) was observed (Fig. 2C). Taken together, the results observed from the simplified ex vivo screening assay reasonably predicted the behavior of a selected panel of non-competitive and competitive strains in a prophylactic in vivo mouse model.

Next, we wanted to assess whether the commensal strain with the highest clearance rate, *E. coli* MR102, can also decolonize an MDR *E. coli* strain after its successful colonization, *i.e.*, in a therapeutic manner. Therefore, we changed the colonization order of *E. coli* strains in the mouse model. Specifically, ampicillin-treated mice were colonized with the MDR1 and then with the commensal MR102 strain four days later. Mice treated with MR102 could reduce the colonization levels of the MDR *E. coli* strain compared to the controls 10- to 100-fold starting at day nine after MR102 administration (Fig. 2D). At day 14, 50% of mice could already clear out MDR1, reaching 80% clearance at day 28 (Fig. 2E). Interestingly, MR102 colonization was still detectable in 6 and 4 animals on days 14 and 28 (up to 10$^9$ CFU/g) respectively, while it was below the level of detection in the rest of the mice (Fig. 2F).

To test whether the prophylactic protective effect is limited to the MDR *E. coli* MDR1, we selected another MDR *E. coli* isolate (NRZ 21236; porin loss, ESBL), here referred to as MDR2, belonging to ST131, which is the most dominant ST linked to nosocomial infections caused by ESBL-carrying *E. coli*[16,53]. This strain was evaluated in the previously described mouse model against *E. coli* MR102. Precolonization with MR102 led to 1000-fold and 100,000-fold reduced colonization levels of MDR2 on days 1 and 6 after colonization, respectively (Fig. 2G). Moreover, MDR2 was completely cleared on day 9 in mice precolonized with *E. coli* MR102 compared to no clearance, even until the end of the experiment, in PBS-treated mice (Fig. 2H). Similarly, even if MDR2 was the first colonizer, MR102 could completely clear MDR2 on day 9 (Fig. S2E, F).

Together, these experiments demonstrated that the commensal *E. coli* MR102 strain showed competitive effects against an *E. coli* ST617 MDR strain in a prophylactic and therapeutic approach and against a clinical isolate of the globally disseminating ST131.

### Protection against MDR *E. coli* colonization is associated with accelerated microbiome recovery and the presence of key species

The composition of microbiota plays a vital role in bacterial competition and contributes to the displacement of MDR-E and intestinal pathogens through cooperative niche exclusion[31,32]. To test whether the effect of MR102 depends on other microbes, germ-free (GF) mice were first colonized with MR102 or left untreated (as a control), followed four days later by colonization with the MDR1 strain. MR102 successfully colonized GF mice at levels of 10$^9$ CFU/g (Fig. S3A) and reduced MDR1 colonization by approximately tenfold (*p* < 0.0001) (Fig. S3B). However, MR102 failed to clear MDR1 within six weeks (Fig. S3B), demonstrating that cooperation with other microbes is essential for the full protective effect. To identify microbial signatures associated with the protective effect of MR102, we longitudinally characterized the microbiota composition from MR102-colonized SPF mice using 16S rRNA amplicon sequencing. As controls, we included untreated mice, mice treated with ampicillin and PBS, as well as MR158-, MR103-, MK192-, and LK192-precolonized mice. All samples were from mice subsequently challenged with *E. coli* MDR1 (*n* = 9−11 mice per group, *n* = 525 samples in total). Longitudinal analysis revealed a dynamic course of microbiota disruption by ampicillin treatment followed by a gradual recovery of the microbiota (Fig. 3A, B, E S3C). Of note, samples taken during ampicillin treatment were dominated by *Enterobacteriaceae* comprising both the commensal and MDR1 strains (Figure S3C). Comparison of

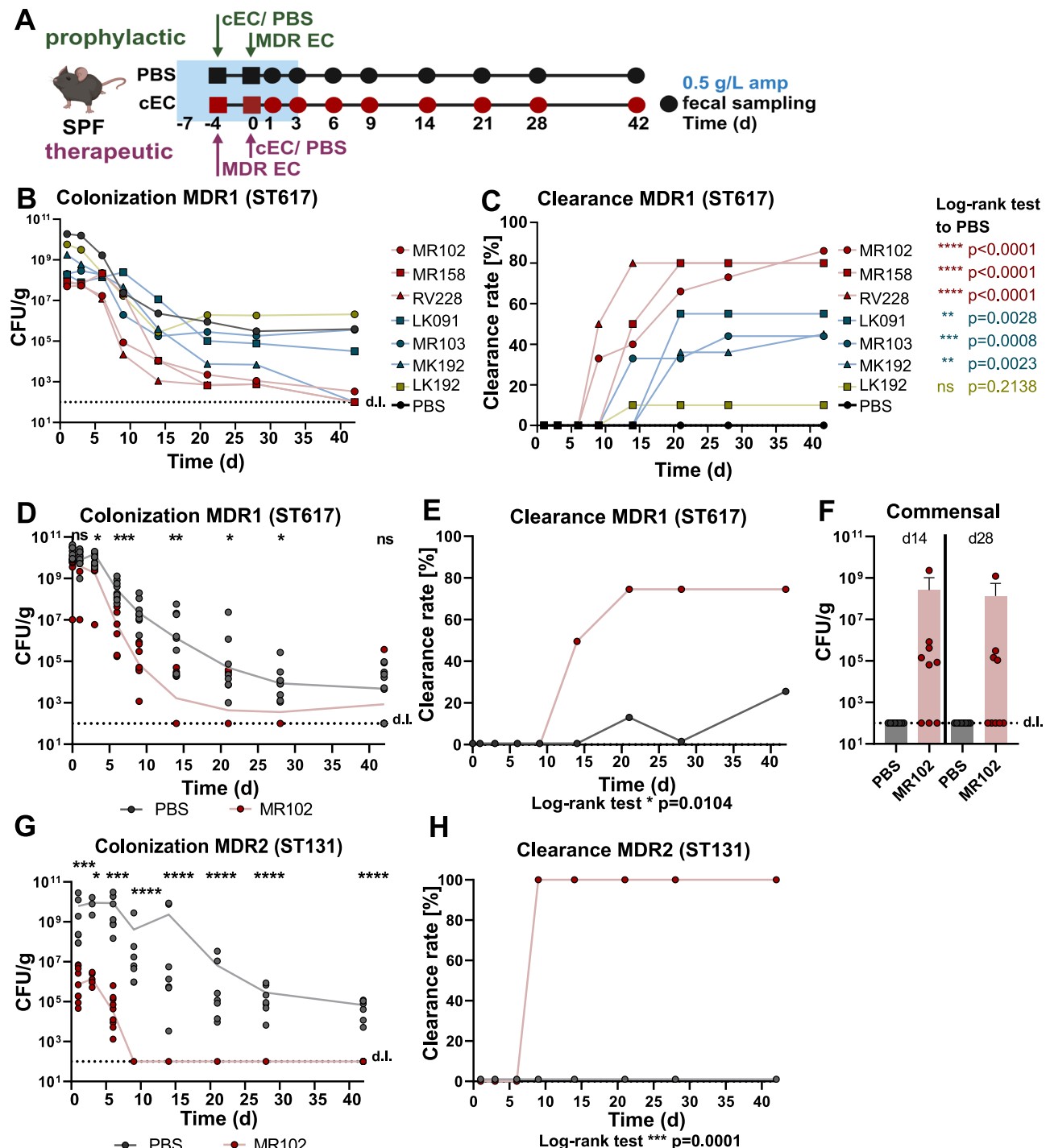

**Fig. 2 | Specific commensal *E. coli* strains enable decolonization of MDR *E. coli* in a preventive and therapeutic manner. A** Workflow of in vivo experiments. For the prophylactic model (**B**, **C**, **G**, **H**), SPF mice were treated with ampicillin in their drinking water three days before oral gavage with commensal *E. coli*/ PBS. After four days, mice were orally gavaged with 10[8] CFUs *E. coli* MDR1 (MHH). For the therapeutic model (**D**–**F**), colonization order was reversed. Fecal samples were taken to monitor fecal colonization levels and 16S rRNA gene sequencing at days 1, 3, 6, 9, 14, 21, 28, and 42. Created with Biorender. **B** Resulting fecal colonization levels of *E. coli* MDR1 after different time points of colonization in the prophylactic model. Data represents the geometric mean and SEM of two to three independent experiments with n = 9–17 mice per group. **C** Clearance kinetics of *E. coli* MDR1 (clearance = CFU/g below the detection limit in feces). *P*-values represent the Log-rank (Mantel-Cox) test with *$p < 0.05$, **$p < 0.01$, ***$p < 0.001$, ****$p < 0.0001$. **D** Resulting fecal colonization levels of *E. coli* MDR1 in the therapeutic model. Geometric mean and SEM of two independent experiments with n = 8 mice per

group. P-values represent the Log-rank (Mantel-Cox) test with *$p < 0.05$, **$p < 0.01$, ***$p < 0.001$, ****$p < 0.0001$. **E** Clearance kinetics of *E. coli* MDR1 after different time points of colonization (clearance = CFU/g below the detection limit in feces). P-values represent the Log-rank (Mantel-Cox) test with *$p < 0.05$. **F** Resulting fecal burden of *E. coli* MR102 after different time points of colonization. CFU/g could be identified by selective plating on MacConkey base supplemented with 10 g/L D-Maltose. Mean and SEM of two independent experiments with n = 8 (PBS) and n = 9 (MR102) mice per group. **G** Resulting fecal burden of *E. coli* MDR2 (NRZ 21236, ST131) after different time points of colonization in the prophylactic model. Two independent experiments with n = 7–9 mice per group. *P*-values indicate a non-parametric Kruskal-Wallis test *$p < 0.05$, **$p < 0.01$, ***$p < 0.001$, ****$p < 0.0001$. **H** Clearance kinetics of *E. coli* MDR2 after different time points of colonization (clearance = CFU/g below the detection limit in feces). P-values represent the Log-rank (Mantel-Cox) test with *$p < 0.05$, **$p < 0.01$, ***$p < 0.001$, ****$p < 0.0001$.

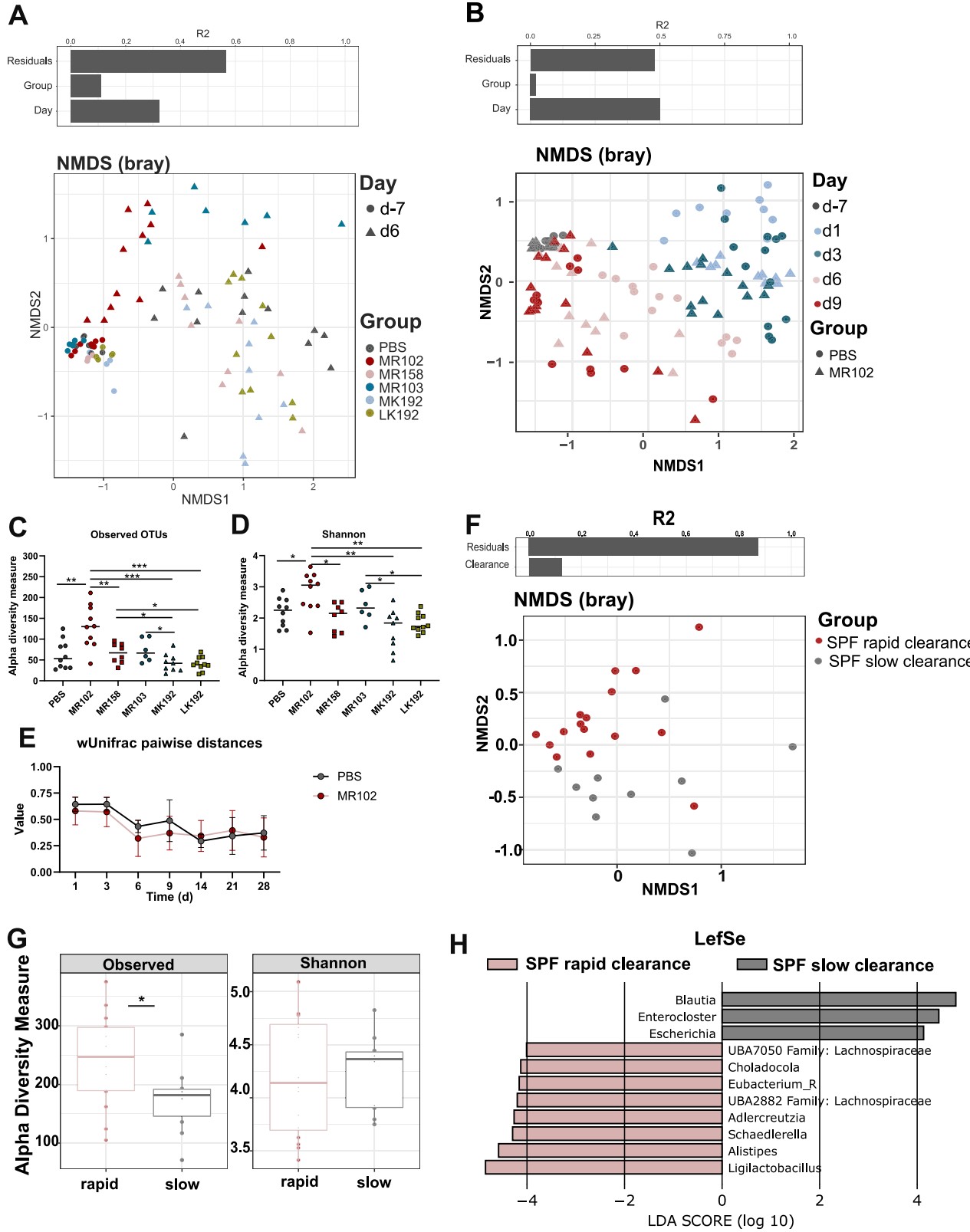

microbiota composition at day 6, which is the first timepoint after stopping the ampicillin treatment, with the untreated samples, showed different clustering when comparing the different treated groups (Fig. 3A). Interestingly, MR102-colonized mice had a significantly higher alpha diversity on day 6 compared to all other groups (Fig. 3C, D). However, no clustering was observed when comparing mice precolonized with either competitive or non-competitive strains.

Since SPF mice colonized with MR102 did not cluster distinctly compared to control mice, we explored an alternative analytical approach. We specifically focused on SPF mice precolonized with MR102 and categorized them based on their varying clearance kinetics

**Fig. 3 | The protective effect is dependent on the microbial context. A** β-diversity of fecal samples of mice treated with PBS, MR102, MR158, MR103, MK192 or LK192 at day-7 or 6 was analyzed using the Bray-Curtis dissimilarity matrix and NMDS. **B** β-diversity of fecal samples of mice treated with PBS or MR102, was analyzed from day −7 to day nine using the Bray-Curtis dissimilarity matrix and NMDS. **C, D** α-diversity represented by observed OTUs or Shannon diversity. *P*-values indicate a nonparametric Kruskal-Wallis test *$p < 0.05$, **$p < 0.01$, ***$p < 0.001$, ****$p < 0.0001$. **E** β-diversity using pairwise wUnifrac distance after antibiotic treatment. P values indicate a nonparametric Kruskal-Wallis test *$p < 0.05$. Mean and SD with $n = 8$ (PBS) and $n = 10$ (MR102) mice per group (**F**) β-diversity of fecal samples of slow and rapid clearance mice at day nine was analyzed using the Bray-Curtis dissimilarity matrix and NMDS. **G** α-diversity on day nine of slow and rapid clearance phenotype mice represented by observed OTUs and Shannon index $n = 16$ rapid-clearer and $n = 10$ slow-clearer in box with whiskers (min to max). **H** Analysis of differentially abundant bacterial species in mice that showed rapid clearance vs. slow clearance phenotype by LefSe.

of MDR1. Mice that cleared MDR1 by day 14 were classified as "rapid clearers", while those that did not clear MDR1 until day 14 were classified as "slow clearers" (Fig. S3D, E). Samples from these mice taken on day 9, the time point preceding the clearance, underwent shotgun metagenome sequencing and subsequent taxonomic analysis. NMDS analysis indicated a separation between "rapid clearers" and "slow clearers" (Fig. 3F). Notably, "rapid clearers" exhibited a higher relative abundance of *Lactobacillaceae* and greater alpha diversity compared to "slow clearers" (Fig. 3G, S3F). Additionally, LefSe analysis demonstrated an enrichment of the genus *Ligalactobacillus*, particularly *Ligalactobacillus murinus*, in rapid clearers (Fig. 3H).

The microbiota of the Oligo-Mouse-Microbiota-12 (OMM[12]) and Oligo-Mouse-Microbiota-19 (OMM[19]) models contain *Lactobacillaceae*, specifically *Ligalactobacillus reuterii* (present in both OMM[12] and OMM[19]) and *L. murinus* (found in OMM[19]). Notably, OMM[12] mice also harbor an *E. coli* strain in its native state. To test whether MR102 could inhibit MDR1 colonization in these defined community models, gnotobiotic mice were first colonized with MR102 and then challenged with MDR1. Importantly, despite the presence of the endogenous *E. coli* strain, *E. coli* MR102 could successfully colonize OMM[19] mice, as well as OMM[12] mice (Fig. S3G, H). MR102 reduced MDR1 colonization in OMM[12] mice by 100-fold and in OMM[19] mice by 10-fold throughout the experiment (Fig. S3I, J); however, no clearance was observed. Next, metagenome sequencing was performed to identify highly resolved taxonomic and functional differences in the microbiota between SPF mice and the OMM[12] ($n = 15$) and OMM[19] ($n = 17$) models. Specifically, we chose fecal samples from SPF mice, control ($n = 16$), and MR102-colonized ($n = 26$) taken on day nine after the challenge with MDR1. The most significant functional differences were observed between microbiota settings and did not depend on precolonization with MR102. OMM[12] and OMM[19] mice clustered distinctively but close to each other. SPF mice formed another cluster displaying broader functional differences between individual mice (Fig. S3K). Collectively, this data demonstrates that MR102 requires cooperation with yet-to-be-defined bacteria to promote the displacement of MDR1 from the gut. *Lactobacillaceae* may play an additive role, but the presence of *L. murinus* strain from OMM[19], together with the other OMM[19] members, is not sufficient to complement MR102 in fully inhibiting MDR1 colonization.

## Protective *E. coli* strains display distinct carbohydrate utilization patterns

Diverse mechanisms, including the production of antimicrobial molecules and nutrient competition, have been identified to promote antagonism between commensal *E. coli* strains and pathogenic *Enterobacteriaceae*[31,32,34,35]. The dependency of *E. coli* MR102 on a complex microbiota to promote gut decolonization suggests that the metabolic properties of competitive bacterial strains may play a key role. We initially tested this presumption via a hypothesis-free genome-wide association analysis (GWAS) to identify genetic variants statistically linked with the in vitro competitive phenotype (*i.e.*, the 10 % of strains with the most potent inhibition). To capture core and accessory genome variation, genetic variants were encoded as unitigs, unambiguous assembled k-mers derived from a de-Brujin graph built from all genome assemblies (see methods for details). Eighty-five

unitigs were identified (Supp. Data 3), passing the association p-value threshold of $3.29 \cdot 10^{-08}$, which were mapped back to all genomes. On average, 6 unitigs per genome were mapped across all phylogroups except for B1, for which, on average, 18 unitigs per genome were mapped (Fig. S4A), indicating that we predominantly identified genetic variants associated with the competitive phenotype in this phylogroup. Overall, the unitigs mapped to 33 gene clusters, 15 of which were core genes, 15 'shell' (frequency between 15% and 95%), and 3 'cloud' (frequency <15%) (Supp. Data 3). Of those 33 gene clusters, six gene clusters were related to direct bacterial antagonism: one related to microcin (*pmbA/tldE*) and five related to type VI secretion systems (T6SS, *tssE*, *tssF*, *tssG*, *vgrG*, and *hcpC*). Additionally, we identified six other gene clusters related to metabolism: *glnE* (regulation of glutamine synthetase), *ubiH* (contributes to ubiquinone synthesis, respiration), *nuoC* (part of the electron transport chain), *btuD_3 ~ ~fruK_2* (part of an unannotated simple sugar transport system), *fnr* (gene regulator involved in adaptation to anaerobic growth), and *paaZ* (catabolism of phenylacetic acid). We also found enrichment in genes belonging to KEGG module M00334 (formaldehyde assimilation, xylulose monophosphate pathway, corrected Fisher's test p-value $2 \cdot 10^{-6}$) (Supp. Data 3). This analysis indicates that genetic variations in gene clusters contributing to bacterial antagonism and various areas of metabolism are associated with the in vitro competitive phenotype. However, despite many analyzed strains, this analysis falls short of identifying prevalent features associated with competitive strains, particularly beyond the B1 phylogroup. The intrinsic significant genetic variance within *E. coli* and the possible contribution of various mechanisms to the quantified phenotype are likely responsible for this limitation.

Since two of the three *E. coli* strains with validated in vivo protective effects (MR102 and RV228) belong to phylogroup D, we built on the results of the GWAS analysis and next systematically evaluated the ability of *E. coli* strains to utilize various carbon sources. Specifically, we quantified the growth of *E. coli* strains tested for their competitive phenotypes in vivo (MR102, MR158, RV228, MR103, LK91, MK192, LK192) and included two MDR strains as control (MDR1 and 2). Growth was assessed aerobically in minimal media (MM9) supplemented with the individual carbon sources ($n = 39$), which included mono-, di-, and oligosaccharides as well as glucosides, sugar alcohols, carboxylic acids, and mucus-derived aminosugars that are frequently present in the gut. As expected, *E. coli* strains differed in the number of carbohydrates they could use and their utilization patterns (Fig. 4A). The competitive strains MR102, MR158, and RV228 utilized 22, 21, and 22 carbon sources, respectively. Non-competitive strains displayed highly variable utilization patterns ranging from 13 (LK192) to 24 (MR103) (Fig. 4A). Similar variations were observed between MDR strains (MDR1: 20, MDR2: 17). Notably, *E. coli* MR102 displayed the most robust and efficient growth and could utilize the most carbon sources ($n = 15$) with a growth area under the curve (AUC) of more than 20 (Fig. S4B). To analyze the utilization patterns, we focused on combinations of previously studied strains in vivo, comparing protective, non-protective, and MDR strains. For instance, MR102, LK192, and MDR1 display significant overlap (15/ 39 tested carbon sources), with all three strains utilizing these carbon sources. Furthermore,

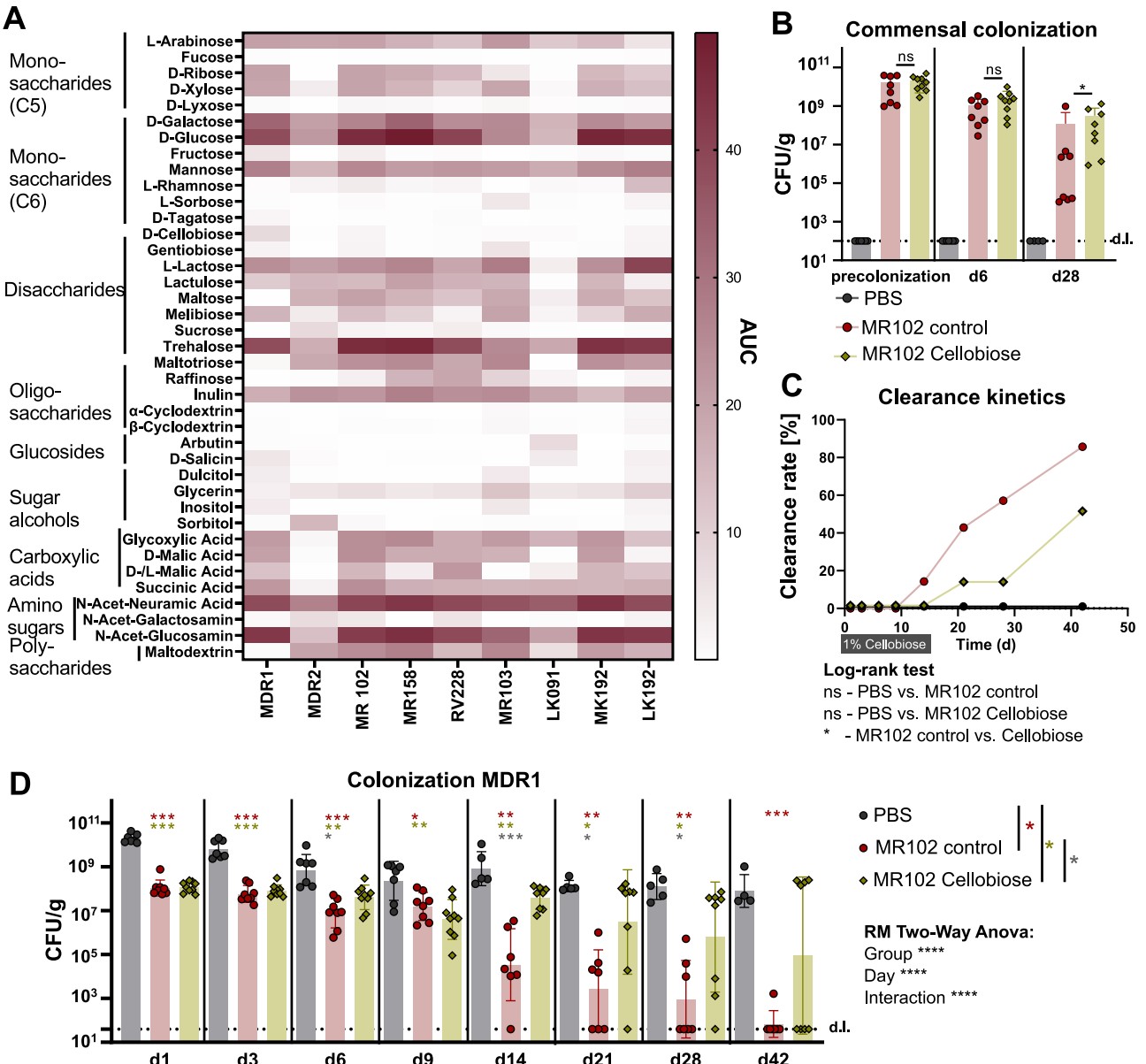

**Fig. 4 | Distinct carbohydrate utilization is involved in intra-species competition. A** The heatmap shows an AUC of 72 h of growth curves of *E. coli* strains in MM9 supplemented with 5 g/L of single carbon sources (*n* = 39). Results of three independent experiments performed in triplicates. **B** Commensal *E. coli* MR102 colonization levels at different time points. Mean and SEM of two independent experiments with *n* = 4 (PBS) and *n* = 8 (MR102 and MR102+Cellobiose) mice per group. *P*-values indicate a nonparametric Kruskal-Wallis test *p* < 0.05, **p* < 0.01, ****p* < 0.001, *****p* < 0.0001. **C** Clearance kinetics of *E. coli* MDR1 after different time points of colonization (clearance = CFU/g below the detection limit in feces). *P*-values represent the Log-rank (Mantel-Cox) test with 0.05. **D** CFU/g of *E. coli* MDR1 of single mice at different time points of colonization. Geometric mean and SEM of two independent experiments with *n* = 4 (PBS) and *n* = 7 (MR102) and *n* = 8 (MR102+Cellobiose) mice per group. mice per group. *P*-values indicate a nonparametric Kruskal-Wallis test *p* < 0.05, **p* < 0.01, ***p* < 0.001, ****p* < 0.0001.

MR102 shows an overlap in utilized carbon sources with MDR1 (D-malic acid, lactulose) and LK192 (maltotriose, maltodextrin, maltose). But, distinct utilization patterns also were evident as certain sugars were exclusively metabolized by the single strains (MDR1: D-cellobiose, D salicin, and D-fructose; MR102: N-acetyl-D-galactosamin; LK192: L-rhamnose) (Fig. S4C).

This difference in carbohydrate utilization between the MDR1 and MR102 strains allowed us to indirectly test whether carbohydrate competition between the strains is relevant in the gut environment[31,35,37], *i.e.*, by supplying mice with cellobiose, which the MDR1 but not the MR102 strain can utilize. Specifically, SPF mice were supplemented with 10 g/l cellobiose in the drinking water, starting three days before precolonization with *E. coli* MR102 until day 14 post-

colonization with MDR1. Otherwise, the experiment was conducted as previously described. Monitoring fecal colonization levels of both bacteria at early time points showed similar colonization levels of *E. coli* MDR1 in mice precolonized with MR102 regardless of cellobiose supplementation. Starting on day 14, colonization levels of MDR1 in mice supplemented with cellobiose were 10-fold higher, reaching a 10,000-fold difference on day 42 (Fig. 4D). Accordingly, a diminished clearance of MDR1 after supplementation with cellobiose and a final clearance rate of 50% in cellobiose-treated mice (80% in non-treated mice) could be observed (Fig. 4C). Of note is that MR102 colonization was also affected after cessation of cellobiose treatment, suggesting secondary effects, e.g., in community structure and available niches (Fig. 4B).

Together, these analyses allowed us to identify unbiasedly that genetic variants in factors involved in bacterial antagonism and metabolism are enriched in strains with in vitro competitive phenotypes and that in vivo supplementation of the carbohydrate cellobiose interferes with the protective effect of MR102, suggesting that carbohydrate competition plays a role in the MR102-mediated decolonization of MDR1.

### *E. coli* MR102 competes with *E. coli* MDR1 for mannose in vivo

To more systemically investigate the competition between MR102 and MDR1, the BIOLOG™ Phenotype PM1 and PM2a plates containing 190 different carbon sources were utilized. The strains were grown in single and co-cultures (ratio 1:1) for 24 h to measure their growth longitudinally (as $OD_{600}$). In the end, both strains were enumerated using selective plating to calculate the competitive index ( = CFU MDR1/CFU MR102). Of the carbon sources that were able to support robust growth of at least one of the strains ($OD_{600}$ after 24 hr > 0.3 for single cultures), the majority of carbon sources (58/63, 92%) were dominated by MR102 (competitive index <1). Moreover, all carbon sources that supported the robust growth of both strains in single cultures ($n = 46$) were dominated by MR102 in co-cultures (competitive index <1). For 19 of them, the competitive index was even below 0.1. The lowest competitive index, meaning the most substantial reduction by MR102, could be observed in the monosaccharides D-mannose, D-xylose, D-ribose, L-rhamnose, D-fructose, the disaccharide D-melibiose, as well as in pyruvic acid, D-glucosamine, and uridine (Fig. 5A). For six carbon sources that supported robust growth of MR102 and MDR1 as well as LK192 (as non-competitive control) the competition assays were repeated in minimal media with defined concentrations of the carbon sources. While co-cultivation with MR102 reduced the competitive index compared to LK192, LK192 did not lead to a competitive index smaller than 0.4, even with its most robust competition in D-melibiose. The most substantial competition between MDR1 and MR102 could be observed in D-galactose, D-mannose, and D-melibiose, with a competitive index smaller than 0.001. Notably, these competitive interactions required oxygen (Fig. 5C).

To test whether competition in vitro and the protective effect of MR102 against MDR1 in vivo is based on the competition for D-mannose, we generated a strain (MR102ΔmanA) lacking *manA*, the mannose-6-phosphate isomerase essential for utilization of mannose, as well as a complementation strain (MR102ΔmanA::manA) (Fig. S5A, B). In vitro, competition assays between MDR1 and MR102 WT, ΔmanA, and ΔmanA::manA in cecum content and minimal media supplemented with mannose revealed a *manA*-dependent effect on competition (Fig. S5C). Next, SPF mice were colonized with MR102 WT, ΔmanA, or ΔmanA::manA and subsequently challenged with the MDR1 strain. From day 1 to day 9, fecal MDR1 colonization levels were 100-fold reduced in the three MR102-precolonized groups compared to the PBS control. Still, no *manA*-dependent effect could be observed during this period (Fig. 5D). However, while MR102 WT- and ΔmanA::manA-precolonized mice cleared out MDR1 until day 14, MR102 ΔmanA-precolonized mice showed a delayed clearance kinetic, with clearance starting at day 14 and 100% clearance at day 21 (Fig. 5F). Of note, *E. coli* MR102 WT, ΔmanA, and ΔmanA::manA were able to colonize mice to the same level at all time points (Fig. 5E). Together, these results demonstrate that competition for mannose contributes to gut decolonization, but other yet unknown factors contribute to efficient gut decolonization by MR102.

### Combinations of probiotic bacteria can broaden the target spectrum

Due to the large genomic variability and metabolic diversity of *E. coli* strains, including between MDR strains, we wanted to assess whether the protective effect demonstrated against strains of ST617 and ST131 can also be observed against MDR *E. coli* strains from other STs. To first investigate the competitive effects of commensal isolates against a variety of strains in vitro, we screened different commensal isolates against a panel of 16 additional MDR *E. coli* strains and one enteropathogenic *E. coli* (EPEC) strain as well as MDR1 and MDR2 (Table S2). These strains were co-cultured (1:10 ratio) individually with two competitive *E. coli* strains from the strain collection (MR102, MR134), or as controls with EcN, LK192 and MR103 (non-competitive strains). Furthermore, we included *K. oxytoca* MK01, which we previously identified as able to metabolize many diverse carbon sources and promote gut decolonization of MDR *K. pneumoniae*[31]. The commensal strains show different competitive effects (Fig. 6A). MK01 could reduce the CFUs of most of the strains (12/19), followed by the strains MR102 (7/19), MR103 (4/19), MR134 (5/19), and LK192 (3/19). The EcN strain could only reduce the CFUs of two MDR strains. Notably, while we could identify one MDR strain (MDR2, NRZ21236, ST131) that was reduced by all of the strains, many of the competitive effects were scattered or even strain-specific, *i.e.*, MDR strains were inhibited by a subset or even a single strain, respectively. Moreover, none of the tested isolates could reduce one MDR strain (NRZ55652, ST167, referred to as MDR3 from hereon). We, therefore, hypothesized that combinations of *E. coli* MR102 with *K. oxytoca* MK01 could broaden the spectrum of competitive effects. Indeed, the combination of MK01 and MR102 could reduce the CFUs of 84% (16/19) of the tested strains; a notable exception was MDR3. In contrast, the combination of MK01 and EcN could reduce the CFUs of only one additional strain compared to the MK01 strain alone (Fig. 6A). Notably, we also observed individual combinations that seem to be less effective in inhibiting MDR-E strains in vitro, which warrants future studies of potential growth promoting interactions. To support our working model of decolonization mediated via niche exclusion, we compared, using Biolog plates PM1 and PM2 (190 carbohydrates in total), the carbohydrate utilization patterns of the *E. coli* strains MR102, EcN, MDR1, MDR2 and MDR3, as well as *K. oxytoca* MK01. Interestingly, MDR3, which could not be inhibited in vitro, and MR102 could utilize more carbohydrates ($n = 52$ and $n = 54$) compared to EcN ($n = 48$), MDR2 ($n = 26$) and MDR1 ($n = 45$), indicating metabolic advantages of MDR3 and MR102. *K. oxytoca* MK01 could utilize the most carbon sources ($n = 82$) (Fig. S6). Combining the carbohydrate utilization profiles of MR102 and MK01, both bacteria cooperatively overlap the carbohydrate utilization profile of MDR3, except for one carbon source (lactulose) (Fig. 6B). Together, these results support the potential of strain combinations for their decolonization properties via niche exclusion, yet, that it may be challenging to model them in vitro.

Therefore, we decided to test the utility of *E. coli* MR102 and *K. oxytoca* MK01 as a potential probiotic mixture and first assessed their potential for coexistence in vivo. Ampicillin-treated SPF mice were co-colonized with both strains and remained colonized at comparable levels with both strains for 6 weeks, demonstrating that these strains do not interfere with each other (Fig. 6C). To next investigate whether co-colonization extends the spectrum of cleared MDR strains, mice were precolonized with either *E. coli* MR102, *K. oxytoca* MK01, or the mixture (Fig. S7A) and then challenged with different MDR strains (10^8 CFUs). We selected a panel of four MDR-E strains, including two *E. coli* strains, MDR1 (as control) and MDR3 (as a challenging competitor strain), as well as an *Enterobacter cloacae* strain and a *K. pneumoniae* (strain MD) strain. Notably, the co-colonization of MR102 and MK01 did not interfere with the decolonization of MDR1 (Fig. 6D). For MDR3, MR102 or MK01 alone were able to reduce colonization levels 5 to 10-fold at days one and three compared to the PBS-treated group, whereas together, they reduced the colonization levels 1000-fold. Moreover, all mice precolonized with MR102 and MK01 had cleared out MDR3 from the gut by day nine, while clearance in the single-colonized groups reached 62% (MK01) and 50% (MR102) at day 14 and 75 % (MK01) and 71 % (MR102) after 6 weeks. (Fig. 6E). For the tested *K. pneumoniae* and *E. cloacae* strains, no statistically significant

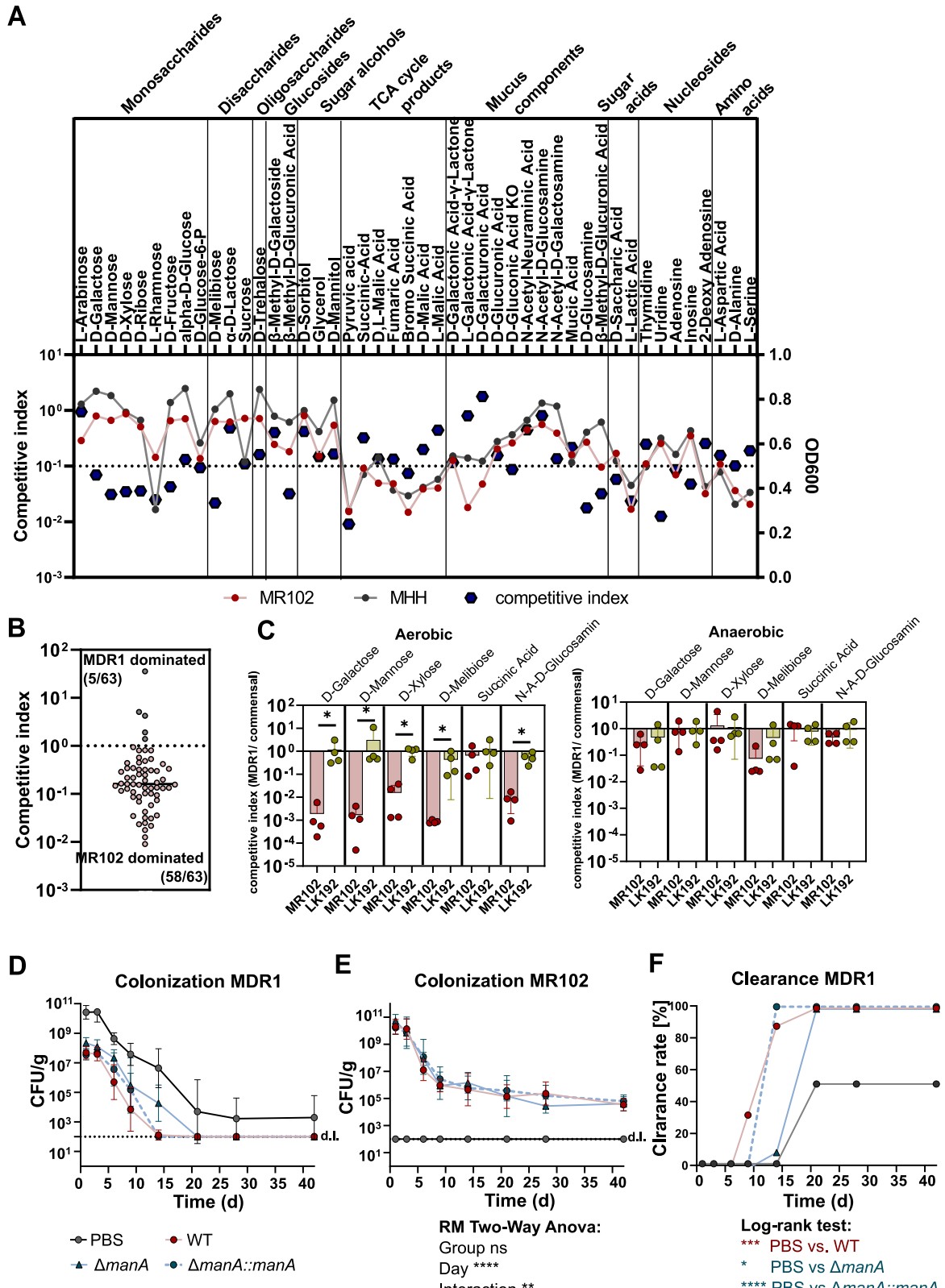

acceleration of clearance was observed for the combination vs. the single strains, as either MK01 or both strains alone already promoted efficient clearance, respectively (Fig. 6F, G). Two recent studies identified an in vivo anti-microbial effect of *K. oxytoca*-produced toxins under specific environmental conditions[36,54]. Therefore we tested the combination of MR102 with the toxin-deficient *K. oxytoca* MK01 *ΔnpsA* strain, but could not observe differences in the decolonization kinetics

(Figure S7L, M). Finally, we reversed the colonization order for the MDR-E strains (*E. coli* MDR1, MDR2; *K. pneumoniae*) and the combination to test a therapeutic set-up. Strikingly, even a single gavage of the combination of MK01 and MR102 was sufficient to achieve 75-80 % clearance within three weeks (Fig. S7F–K). These results demonstrate that a combination of candidate probiotic bacteria can promote efficient gut decolonization of the tested clinical MDR bacteria in a

**Fig. 5 | Protective *E. coli* strain is superior to MDR *E. coli* in direct competition for specific carbohydrates. A** Results of competition assay in Biolog® Phenotypic microarray competitive index (CFU/ml *E. coli* MDR1/ CFU/ml *E. coli* MR102) and OD$_{600}$ of the respective strains of all carbon sources that both strains could utilize. One dot represents data from three independent experiments. **B** Results of competition assay in Biolog® Phenotypic microarray competitive index (*E. coli* MDR1/ *E. coli* MR102) of all carbon sources that could be utilized by at least one of the strains. One dot represents data from three independent experiments. **C** Competition assay in MM9 supplemented with 5 g/L of the respective carbon source under aerobic and anaerobic conditions. P values indicate a nonparametric Kruskal-Wallis test *$p < 0.05$. Mean and SD with one dot representing data from one experiment performed in duplicates or triplicates. **D** Resulting fecal burden of *E. coli* MDR1 after different time points of colonization. Geometric mean and SD of two independent experiments with $n = 8$ (PBS), $n = 9$ (WT) and $n = 10$ (*ΔmanA* and *manA::manA*) mice per group. **E** Commensal *E. coli* MR102 colonization levels at different time points. Geometric mean and SD of two independent experiments withof two independent experiments with $n = 8$ (PBS), $n = 9$ (WT) and n = 10 (*ΔmanA* and *manA::manA*). **F** Clearance kinetics of *E. coli* MDR1 after different time points of colonization (clearance = CFU/g below the detection limit in feces). *P*-values represent the Log-rank (Mantel-Cox) test with 0.05, **$p < 0.01$, ***$p < 0.001$, ****$p < 0.0001$.

prophylactic and therapeutic setting, including strains from clinically relevant *E. coli* STs and other MDR-E.

## Discussion

Members of the group of Enterobacterales are in constant competition for their metabolic niche in the gut environment, within species, between closely related species, and the surrounding microbiota[31–34]. This competition represents a promising approach for developing novel therapeutic agents that promote gut decolonization of multi-drug resistant Enterobacterales (MDR-E). While this approach would not directly replace the need to develop innovative antimicrobials to fight systemic infections, it would aid in the ongoing fight against MDR bacteria, given that gut colonization frequently serves as a first step into the progression toward an infection[7]. One particularly challenging MDR bacterial species is *E. coli*, characterized by a high intra-species diversity and a large pangenome, resulting in sizeable metabolic flexibility when considering individual strains and the species[55]. Past studies identified competitive effects of commensal *E. coli* strains against their pathogenic counterparts[43,56], albeit predominantly focusing on single strains and strain-specific characteristics. However, recent studies of more extensive strain collections of commensal bacteria identified different promising functional properties only found in subsets of strains[57,58]. Hence, we isolated a comprehensive collection of commensal *E. coli* strains ($n = 439$) to characterize their potential to inhibit the growth of a representative indicator MDR *E. coli* strain. Ex vivo co-cultivation unveiled distinct competitive effects of respective strains with in vitro inhibitory strains being enriched in specific *E. coli* phylotypes, specifically B1 and D. Still, this side observation would need further experimental validation, whether it is also occurring in other models or with other MDR *E. coli* strains. Intriguingly, the probiotic *E. coli* Nissle 1917 (EcN) did not belong to the competitive isolates, even though it was previously described to protect against pathogenic *E. coli* colonization[43,44,59]. Subsequent validation of the competitive effects in a mouse model with antibiotic-disrupted SPF microbiota supported the potential of the ex vivo assay to predict competitive effects. The tested competitive strains enabled the complete decolonization of MDR *E. coli*. In contrast, precolonization with non-competitive strains resulted in lower clearance rates, albeit with similar abilities to colonize the gut in the selected mouse model. We conclude, that the ex vivo screening assay offers a valuable opportunity for initial screenings of potential probiotic candidates, even though it is indispensable to further evaluate these results as this model is still an artificial set up. Together, this demonstrates that the combination of in vitro and in vivo model systems allows the screening of strain-specific functional differences that could be explored to identify promising properties without prior knowledge of desired strain properties.

Various microbes with distinct metabolic properties colonize the human gastrointestinal tract and invading bacteria need to be able to explore specific metabolic niches. Thus, metabolic attributes of resident bacteria are recognized as critical modulators of competition, and specifically, competition for carbohydrates among Enterobacterales is the foundation of the Freter nutrient-niche hypothesis[38,60].

Additionally, the dynamics of Enterobacterales competition are highly influenced by the microbial context, with examples of overall diversity or critical species further modulating and restricting available niches and growth-limiting nutrients[31,32,34,40]. Consistent with these observations, commensal *E. coli* MR102 could provide full clearance exclusively in SPF mice but not in gnotobiotic or germ-free (GF) mice. Notably, colonization with *E. coli* MR102 led to an accelerated microbiota recovery after antibiotic-induced dysbiosis, which could further contribute to the successful clearance of *E. coli* MDR1. Clearance was associated with the presence of a *L. murinus* strain from the SPF microbiota. The OMM[19] microbiota, for which no clearance was observed, also harbors a *L. murinus* strain. This further underlines our hypothesis of strain-specific competitive effects and needs further experimental evaluation. Yet, growth inhibition in the in vitro assay and the MR102-mediated a 10- to 100-fold reduction of MDR1 colonization in GF, OMM[12], and OMM[19] mice indicated that the competition is mediated through multiple mechanisms. A GWAS analysis identified in phylogroup B1 competitive isolates genes associated with metabolism and direct bacterial antagonism, such as type VI secretion systems. Due to the large intra-species genetic diversity and the higher number of phylogroup B1 competitive isolates, this approach identified more candidate genes in the B1 isolates than phylogroup D isolates. Therefore, we complemented the GWAS approach through functional assays, and selected strains were analyzed regarding their ability to metabolize various carbon sources. This revealed distinct carbohydrate utilization patterns among MDR and commensal strains with different competitive effects. Moreover, competition assays conducted in minimal media supplemented with single carbon sources demonstrated the ability of protective *E. coli* MR102 to dominate MDR *E. coli* across the majority of tested carbon sources. Mainly, co-cultivation in monosaccharides led to a robust reduction of MDR *E. coli* CFUs, suggesting that robust growth across various carbon sources is associated with favorable properties. In vivo competition of *E. coli* MR102 *ΔmanA*, lacking the ability to utilize mannose as a sole carbon source, resulted in a diminished protective effect compared to *E. coli* MR102 WT. Together with the ability of cellobiose to reduce the gut clearance of the MDR1 *E. coli* strains, this suggests that metabolic competition contributes to gut decolonization. However, additional factors are likely contributing to the complex phenotype.

The strategic design of effective probiotics to promote gut decolonization of MDR-E requires the comprehensive evaluation of the target spectrum of such probiotic candidates. We screened a panel of 18 MDR strains representative of various relevant sequence types (STs) to elucidate the target spectrum of potential probiotic candidates. While *E. coli* MR102 displayed favorable properties against a subset of strains, including against ST131, a significant threat in the nosocomial setting, other STs were not as well targeted. However, following the guiding principle of niche restriction to displace unwanted strains, combining the protective *E. coli* with *K. oxytoca* MK01, a metabolically versatile probiotic candidate, expanded the target spectrum compared to the single strains, including a strain of *E. coli* ST167. Notably, overall, more MDR *E. coli* strains could be reduced by the combinations. However, since we demonstrated some examples of interference

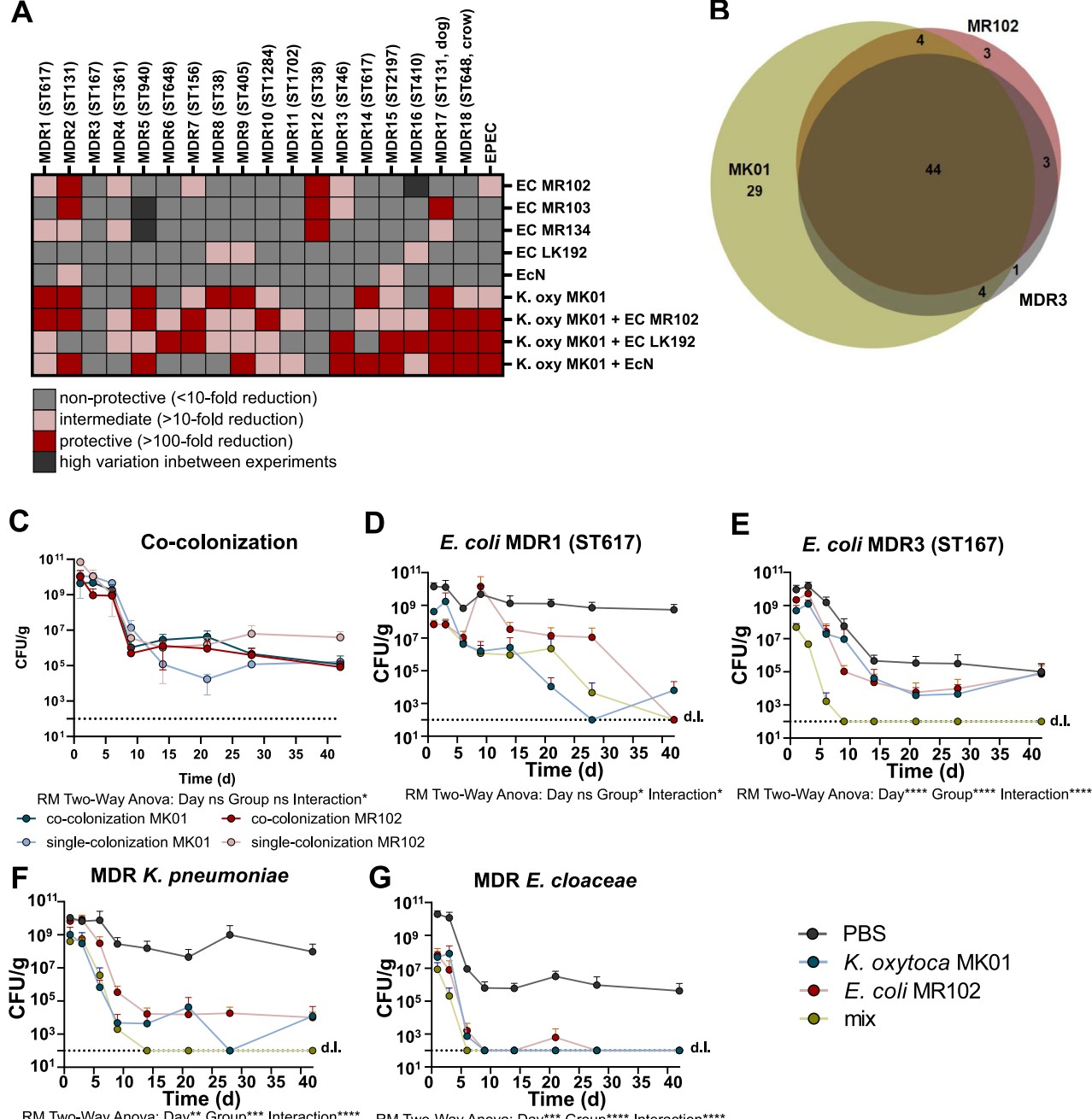

**Fig. 6 | A Combination of metabolically diverse *Enterobacteriaceae* can enlarge the target spectrum of MDR-E. A** Heatmap summarizing the results of an ex vivo competition assay of five commensal *E. coli* strains, *K. oxytoca*, and combinations of *E. coli* and *K. oxytoca* against a panel of 18 MDR *E. coli* strains and one EPEC strain. Fold reduction of CFU/ ml in co-cultures to MDR *E. coli* control is classified as non-protective (100-fold reduction). **B** Venn diagrams displaying carbon source overlap of *E. coli* MDR3 (NRZ 55652, ST167), *K. oxytoca* MK01, and *E. coli* MR102 or EcN. **C** Resulting fecal burden of *E. coli* MR102 and *K. oxytoca* MK01 in co-colonization

after different time points of colonization. Geometric mean and SD of one experiment with n = 5 (co-colonization MR102 and MK01, n = 4 (single-colonization MR102 and n = 3 (single-colonization MK01) mice per group. **D–G** Resulting fecal burden of MDR-E after different time points of colonization. Geometric mean and SD of two independent experiments with n = 9 (**D**), n = 8 (PBS, MK01, MR102), n = 10 (mix) (**E**), n = 8 (PBS, MK01), n = 9 (MR102, mix) (**F**), n = 9 (PBS, MK01), n = 8 (MR102) and n = 10 (mix) (**G**) mice per group.

with the competitive effect in these combinations, this needs to be analyzed in a more comprehensive and systematic manner. Varying levels of gut colonization for the different MDR-E strains was observed, still all tested strains showed sufficient gut colonization for the experimental evaluation of MDR-E decolonization. Further testing will be required to identify the full spectrum of protection, i.e., additional representatives from the ESKAPE panel.

Future probiotic strains should not only have beneficial effects but should also be safe for the host. For instance, EcN was previously reported to encode colibactin, a genotoxin associated with colorectal cancer[46,47]. Consistent with previous studies, we found that colibactin was only encoded by phylogroup B2 strains from our strain collection[51,52]. Interestingly, the phylogroup B2 generally encoded a higher number of virulence genes than other phylogroups. Notably,

antimicrobial resistance genes (ARGs) were evenly distributed through all phylogroups.

Moreover, even though strains were isolated from non-hospitalized donors, the presence of virulence factors or the possibility of acquiring ARGs or VAGs can not be excluded and needs further studying. Thus, additional studies will be required to define on a per-strain basis which potential virulence factors contribute to the protective phenotype and which represent the biggest clinical threat and should be thus excluded. Our findings highlight the metabolic interplay between bacterial species in a microbial community and the microbiota's potential in exploring potential therapeutic or prophylactic live biotherapeutic products in the fight against AMR dissemination and infection.

## Methods

### Contact for reagent and resource sharing
Further information and requests for resources and reagents should be directed to and will be fulfilled by the Lead Contact, Prof. Dr. Till Strowig (till.strowig@helmholtz-hzi.de).

### Ethics statement
All animal experiments have been performed in agreement with the guidelines of the Helmholtz Center for Infection Research, Braunschweig, Germany, the National Animal Protection Law [Tierschutzgesetz (TierSchG) and Animal Experiment Regulations (Tierschutz-Versuchstierverordnung (TierSchVersV)], and the recommendations of the Federation of European Laboratory Animal Science Association (FELASA). The study was approved by the Lower Saxony State Office for Nature, Environment and Consumer Protection (LAVES), Oldenburg, Lower Saxony, Germany; permit No. 33.19-42502-04-21/3817.

### Human cohorts
Human sample and data collections have been performed in agreement with the guidelines of the Helmholtz Center for Infection Research, Braunschweig, Germany, the Ethics Committee Lower Saxony (permit No. 8629_BO_K_2019; No. 8750_BO_K_2019), and the European Data Protection Laws (Europäische Datenschutz-Grundverordnung DSGVO). All human donors have signed a letter of informed consent by the World Medical Association Declaration of Helsinki (Version 2013).

### Mice
All animals used for in vivo experiments were gender and age-matched female and male mice with an age of 8–16 weeks and a weight of 16–30 g at the beginning of the experiment. Mice were kept under a 12 h light cycle (lights on from 7 am to 7 pm) and fed with a chow diet (ssniff catalogue-Nr. V1534-300). C57BL/6 N mice were bred at the animal facility of the Helmholtz Centre of Infection Research. SPF mice were bred under specific pathogen-free (SPF) conditions (Stehr et al. 2009), and germ-free C57BL/N6Tac mice were bred in isolators (Getinge). OMM[12] and OMM[19] mice were generated by colonization with OMM communities. Mice were killed by euthanization with $CO_2$ and cervical dislocation.

### Bacterial strains
Multidrug-resistant (OXA-48 carbapenemse) E. coli MDR1 (2365332, ST617), obtained from the Hannover Medical School and isolated from a rectal swab of a patient, was used for most in vivo colonization experiments and in vitro assays. Multidrug-resistant E. coli MDR2 (NRZ-21236) and MDR3 (NRZ-55652), obtained from the National Reference Center for Multidrug-resistant Gram-negative Bacteria (Bochum) was used for two in vivo colonization experiments. Further commensal E. coli isolates used for in vivo and in vitro competition experiments were isolated from the human cohort studies at the HZI (MikroResist=MR, MikroKids=MK, LöwenKids=LK, RheumaVor=RV). Further information regarding strains can be found in Table S1/ Supp. Data 1).

### Strain isolation
Fresh stool samples were homogenized by bead-beating in a Mini-Beadbeater (BioSpec) with 1 mm zirconia/ silica beads in 1 ml PBS for 50 s. Stored fecal samples were directly taken from glycerol stock. All samples were serially diluted and plated on CHROMagar Orientation and MacConkey agar plates. Colonies were isolated and identified by 16S-PCR and whole genome sequencing.

### Quantification of E. coli colony-forming units
Samples were spread on plates to determine the CFU of bacteria from culture or feces samples under different conditions. Fresh fecal samples were collected and weighted for quantification of CFUs from fecal samples. Afterward, samples were prepared by adding 1 mm Zirconia beads and 1 ml PBS. Samples were homogenized by bead-beating for 50 s (Mini-Beadbeater, BioSpec). To determine CFUs, 25 µl of serial dilutions of samples were plated on LB plates with respective antibiotics (ciprofloxacine, selection for MDR1-3) or MacConkey agar supplemented with Maltose (selection for MR102). Plates were cultivated overnight at 37 °C under aerobic conditions before counting. Colony counts were normalized to plated volume and/ or weight of feces.

### Ex vivo competition in germ-free cecum content
Germ-free mice were sacrificed, and cecal contents were isolated and diluted 1:1 in PBS. Bacteria were grown in 5 ml LB for 16 h at 37 °C under aerobic conditions before adjusting the $OD_{600}$. MDR E. coli strains were adjusted to $OD_{600}$ of 1, and the commensal strains were adjusted to $OD_{600}$ of 0.2. The assay was performed in 96-well plates with 250 µl diluted cecum content with co-cultures of 20 µl of commensal and 10 µl of MDR E. coli strain. Single culturing of the MDR E. coli strain served as a control.

### In vitro competition for carbohydrates
To investigate direct competition or specific carbohydrates, bacteria were co-cultivated in MM9 media supplemented with 5 g/L of the respective carbon source. To do so, bacterial strains were cultivated on R2A agar plates (Difco) overnight, and bacteria were adjusted to an $OD_{600}$ of 0.2 in PBS. 99 µl of MM9 media with the respective carbon source was filled into a 96-well plate, and 1 µl of each strain was added. Co-cultures were cultivated under aerobic conditions at 37 °C for 24 h. After incubation, CFUs were determined as previously described.

### Phenotypic microarrays
For carbohydrate utilization screenings, the phenotypic microarrays PM1 and PM2a from Biolog[TM] were used. Bacterial strains were cultivated on R2A agar plates (Difco) overnight. Colonies were scraped and adjusted to an $OD_{600}$ of 0.2 in PBS. Each well of the PM1 and PM2a plates was filled with 99 µl of MM9 without carbon sources and inoculated with 1 µl of bacterial suspension. For competition experiments, each well was inoculated with 1 µl of bacterial suspension of each strain. The plates were incubated aerobically in a microplate spectrophotometer with continuous shaking, $OD_{600}$ was measured hourly for 24 h. Mixtures were plated on selective agar plates as previously described.

### Growth analysis
Bacterial strains were cultivated on R2A agar plates (Difco) overnight. Colonies were scraped and adjusted to $OD_{600}$ of 0.2 in PBS. Subsequently, a flat-bottomed 96-well plate (Corning®) was filled with 99 µL of MM9 medium supplemented with the respective carbon source and inoculated with 1 µL of the bacterial suspension. Sterile media served as a negative control. Plates were incubated in a microplate spectrophotometer with continuous shaking (BioTek LogPhase 600 Microbiology Reader). $OD_{600}$ was measured hourly over a time course of 72 hours.

### In vivo E. coli colonization

SPF mice were treated with ampicillin (0.5 g/l) 4 d before starting the experiment. OMM[12] and OMM[19] mice did not receive ampicillin. On the day of colonization, respective bacteria were cultivated in 25 ml LB media (1:25) at 37 °C for 3–4 h. Afterwards, the culture was centrifuged at 500 g for 15 min. The pellet was suspended in 10 ml of PBS and the required amounts were calculated. Mice were inoculated by oral gavage of 10[8] CFUs of E. coli diluted in 200 μl PBS. The body weight of the mice was monitored, and feces were collected at different time points (1, 3, 6, 9, 14, 21, 28, and 42 days after gavage) to measure fecal E. coli burden by plating and 16S rRNA sequencing.

### In vivo decolonization experiments

SPF mice were treated with ampicillin (0.5 g/l) 4 d before starting the experiment. OMM[12] and OMM[19] mice did not receive ampicillin. On the day of colonization, bacterial cultures were prepared as described previously. Mice were precolonized by oral gavage with 10[8] CFUs/ 200 μl in PBS. 4 days after precolonization fecal CFUs of respective E. coli strains were checked by plating on selective agar plates. After successful precolonization (CFU/g of at least 10[9] CFUs), the competing E. coli/ MDR-E strain was administered by oral gavage (10[8] CFUs in 200 μl PBS). The body weight of the mice was monitored and feces were collected at different time points (1, 3, 6, 9, 14, 21, 28, and 42 days after gavage) to measure fecal E. coli burden by plating and 16S rRNA sequencing.

### DNA isolation

The microbial community 16S rRNA gene DNA was extracted using precipitation with phenol-chloroform. 0.1 mm Zirconia beads were added to the feces sample until the bottom part of the tube was covered. Afterwards, 500 μl of 2x Buffer A and 200 μl of 20% SDS were added to each sample. Then 500 μl of the bottom phase of Phenol: Chloroform: IAA was added. This mixture was shaken by a bead-beater for 2 min, then cooled down at 4 °C for 2 min, and then again shaken for 2 min. For phase separation, samples were centrifuged at 8000 xg for 5 min at 4 °C. The aqueous phase was transferred to a fresh 1.5 ml tube and 600 μl of Phenol:Chloroform: IAA was added and mixed by inverting samples several times. Then, samples were centrifuged at 12700 xg for 5 min at 4 °C for the second phase separation. Again, the aqueous phase was transferred to a fresh 1.5 ml tube. 600 μl of 100% Isopropanol (stored at −20 °C) and 60 μl (1/10 volume) of 3 M NaOAc (pH = 5.5) were added. The mixture was vortexed and then stored at −20 °C for at least 1 h or overnight. After incubation at −20 °C, samples were centrifuged at 12,700 x g for 20 min at 4 °C. Then, all liquid was sucked off. The remaining pellet was washed with 1 ml of 70% EtOH (RT) and centrifuged at 12,700 x g for 5 min at 4 °C. Afterwards all liquid was sucked off carefully and the pellet was dried in a speedvac machine for 15 min using no heat. The pellet was resuspended in 100 μl TE-buffer and incubated at 50 °C for 30 min on a shaker at 1000 xg. When the pellet was completely dissolved, 1 μl of 10 μg/ ml RNAse was added. Crude DNA was column purified (BioBasic Inc.) to remove PCR inhibitors.

### 16S rRNA gene amplification and sequencing

16S rRNA gene amplification and sequencing. 16S rRNA gene amplification of the V4 region (F515_GTGCCAGCMGCCGCGGTAA/R806_ GGACTACHVGGGTWTCTAAT) was performed according to an established protocol previously described[61]. Briefly, DNA was normalized to 25 ng/μl and used for sequencing PCR with unique 12-base Golary barcodes incorporated via specific primers (obtained from Sigma). PCR was performed using Q5 polymerase (NewEnglandBiolabs) in triplicates for each sample, using PCR conditions of initial denaturation for 30 s at 98 °C, followed by 25 cycles (10 s at 98 °C, 20 s at 55 °C and 20 s at 72 °C). After pooling and normalization to 10 nM, PCR amplicons were sequenced on an Illumina MiSeq platform via 300 bp paired-end sequencing (PE300). Using the Usearch (v11.0.667) software

package (http://www.drive5.com/usearch/) the resulting reads were assembled, filtered and clustered. Sequences were filtered for low-quality reads and binned based on sample-specific barcodes using QIIME v1.8.0[62]. Merging was performed using -fastq_mergepairs−with fastq_maxdiffs 30. Quality filtering was conducted with fastq_filter (-fastq_maxee 1), using a minimum read length of 300 bp and a minimum number of reads per sample = 1,000. Reads were clustered into 97% identity operational taxonomic units (OTUs) by de-novo OTU picking, and representative sequences were determined by use of UPARSE algorithm of Usearch[63]. Abundance filtering (OTUs cluster >0.5%) and taxonomic classification were performed using the RDP Classifier (Ribosomal Database Project naive Bayesian classifier v2.10.19) executed at 80% bootstrap confidence cut-off[64]. Sequences without matching reference datasets were assembled as de novo using UCLUST. Phylogenetic relationships between OTUs were determined using FastTree to the PyNAST alignment[65]. The resulting OTU absolute abundance table and mapping file were used for statistical analyses and data visualization in the R statistical programming environment package phyloseq[66].

### Shotgun metagenome sequencing

Metagenomic libraries were prepared using the Illumina DNA PCR-Free Library Kit and IDT for Illumina DNA/RNA UD Indexes with previously isolated community DNA. Library preparation followed by the manu[66] facturer's protocol and quantification of library concentrations was performed using the Qubit ssDNA Assay Kit, followed by additional quantification with the KAPA Library Quantification Kit for Illumina. Sequencing was carried out on the NovaSeq S4 PE150 platform with a depth of 25 million reads per sample. Read were trimmed for low adaptor sequences and low quality and filtered for phiX and mouse host reads using the bbmap-software-tools[67]. The sequencing output was analyzed using a mouse gut metagenome catalog (iMGMC) developed in our group[68]. Briefly, reads were mapped to a catalog of 1296 metagenome-assembled genomes (MAGs) for taxonomic annotation, while a collection of 4.6 million unique genes was used for functional annotation. The resulting absolute abundance were normalized to TPM, in addition functional Kegg functional orthologs were interfere to pathways via MinPath[69]. Resulting out-table and mapping file were utilized for statistical analyses and data visualization in the R statistical programming environment package phyloseq.

### Whole genome sequencing

To assess the taxonomy of all E. coli isolates, bacteria were sent for whole genome sequencing. First, genomic DNA was extracted using the ZymoBIOMICS 96 MagBead DNA Kit according to the manufacturer's instructions. Afterwards, libraries of each isolate were prepared using the NEBNext® Ultra™ DNA Library Prep Kit for Illumina®. DNA was fragmented, ligated to adaptors, enriched, and assessed on a Bioanalyzer. Afterwards, libraries of each isolate were prepared using the Illumina DNA PCR-Free Prep and quantified with KAPA Library Quantification Kit Illumina Platforms. Samples were pooled and sent for whole genome sequencing performed by the group of Genome analysis at Helmholtz-HZI Center for Infection Research using NovaSeq 6000 S4 Reagent Kit v1.5 (300 cycles) and targeting depth of 1 million reads per sample. The resulting E. coli genomes have been deposited in GenBank (see Data availability).

### Identification of virulence-associated genes (VAGs), AMR, and colibactin genome islands

We identified genes related to virulence and antimicrobial resistance by running the abritamr tool (v1.0.14)[70] on the nucleotide sequences of the isolates. In order to compare the commensal isolates from this study to pathogenic ones, we performed the same analysis on a collection of 912 clinical E. coli isolates from patients with bloodstream infections[48]. We further identified isolates carrying the colibactin

genome island (*pks* + ) following a similar approach developed previously[71]. We aligned the proteomes of all isolates against the protein sequences of the clb genes encoded in *E. coli* strain IHE3034 (GenBank accession AM229678.1), using Blast+ v2.12.0[72] with an e-value threshold of 1·10-4, a query and subject coverage > 80% and protein sequence identity > 70%. We flagged an isolate to be *pks*+ if we found an homolog of 16 of the 18 clb genes and if the average nucleotide distance between genes encoded in the same contig was below 5kbp.Association analysis

To facilitate the annotation of association hits, we included 8 *E. coli* closed reference genomes from RefSeq, which were downloaded using the NCBI-genome-download package v0.3.3. We included the following genomes: 536 (GCF_000013305.1), CFT073 (GCF_000007445.1), ED1a (GCF_000026305.1), IAI1 (GCF_000026265.1), IAI39 (GCF_000026345.1), MG1655 (GCF_000005845.2), UMN026 (GCF_000026325.1), and UTI89 (GCF_000013265.1). We assigned each genome sequence type (ST) and Escherichia phylogroup using mlst v2.16[73] and ClermontTyping (commit 740c59c)[74], respectively. We generated a de Bruijn graph from all input genomes and extracted unitigs and their presence/absence patterns across all samples using unitig-counter v1.1.0[75,76]. Each unitig with allele frequency <1 encodes for a genetic variant, which can be a large one such as a gene transferred horizontally, as well as short variants such as single nucleotide polymorphisms (SNPs).

To run associations between variants and the protective phenotype, a kinship matrix is required; this can be derived from a core genome alignment, which we generated by first computing the pangenome with panaroo v1.3.0[77], using the "--clean-mode strict" argument, followed by a concatenation of individual alignments for each nucleotide coding sequence belonging to core genes (i.e., with frequency >= 95%). We then used snp-sites v2.5.1[78] and bcftools v1.13[79] to convert the complete core genome alignment to a VCF file containing the variant sites with minimum allele frequency (MAF) > 1% and used this file to derive the kinship matrix using a Python script.

We estimated narrow-sense heritability for the protective phenotype using two covariance matrices: one built from the STs of each strain ("lineage") and another using the kinship matrix described above. We used Limix v3.04[80], assuming normal errors for the point estimate, and we computed the 95% confidence intervals using the ALBI package (commit 90d819e)[81].

We then tested the association between each unitig with MAF > 1% and the protective phenotype using a linear mixed model implemented in pyseer v1.3.6[82]. We determined an appropriate significance threshold by counting the number of unique unitigs presence/absence patterns tested, which reduces the risk of excessively deflating association p-values. We mapped the unitigs passing the significance threshold back to all input genomes using bwa v0.7.17[83] and bedtools v2.31.1[84,85], using the output of panaroo to assign each unitig to a gene cluster. The unitigs were further filtered to reduce the number of spurious associations: unitigs were excluded if they were shorter than 30 bp, if they were mapped to multiple locations in each genome, if they mapped to less than nine samples ( ~ 2% of the sample size) and if they were mapped to more than ten different genes across all samples. We further computed the odds ratio for each gene cluster using the average frequency and effect size as input for a previously developed method for binary phenotypes[86].

We further annotated the gene families with mapped unitigs by taking a representative protein sequence from the genomes encoding each gene family, prioritizing the eight closed reference genomes, and using them as an input for eggnog-mapper v2.1.3[87]. The derived annotations include COG categories, GO terms, and mapping to KEGG entries; we tested an enrichment for each of these annotation systems using the associated gene families as foreground and the annotation for *E. coli* IAI39 as the background, running a Fisher's exact test for each annotation item, and we used an FDR-corrected p-value threshold of 0.05 to indicate annotation items enriched in the associated gene families.

We generated a phylogenetic tree of all samples using the concatenated core genome alignment as an input for FastTree v2.1.11[65,88], using the GTR + CAT model. The tree and associated metadata were visualized using the Microreact web interface[89].

### Deletion mutants

For constructing Δ*manA* deletion mutants, we used a previously adapted two plasmid CRISPR/Cas9 and lambda Red recombination-based genome editing protocol[90,91]. First, *E. coli* MR102 was transformed with pCas. Lambda Red genes were induced with 0.2% L-arabinose for two h. Cells were then co-transformed with a spacer introduced-pSG and linear dsDNA serving as a repair template, assembled from homology arms (500 bp up- and downstream of *manA* gene). After incubation for 24 h at 30 °C on plates supplemented with 200 mg/ml hygromycin and 75 mg/ml spectinomycin, single colonies were screened for the mutant genotype by colony PCR. To reintroduce the previously deleted GOI into the bacterial chromosome, repair templates for complementation were assembled. The RT contains the sequence of the deleted GOI flanked by two HAs. Bookmark complementation requires the introduction of a spacer into the chromosome during gene deletion, which will be targeted during complementation[91,92]. The spacer sequence and a PAM site were synthesized as part of primers designed for HA amplification and assembled via overlap extension PCR.

### Statistical analysis

Experimental results were analyzed for statistical significance using GraphPad Prism v9.1 (GraphPad Software Inc.). Differences were analyzed by Mann-Whitney, Kruskal-Wallis, log-rank test, or one-way ANOVA with various post-hoc tests indicated in each figure legend. The indicated P-values were calculated using a non-parametric Mann-Whitney U or Kruskal-Wallis test to compare totals between groups[93]. OTUs with the Kruskal-Wallis test <0.05 were considered for analysis. P-values lower than 0.05 were considered significant: $*p < 0.05$, $**p < 0.01$, $***p < 0.001$, $****p < 0.0001$. Description of each statistical test and exact p-values are provided in the source data file for each figure item.

### Declaration of generative AI and AI-assisted technologies in the writing process

During the preparation of this work, the authors used Grammarly in order to edit the text. After using this tool, the author(s) reviewed and edited the content as needed and take full responsibility for the content of the publication.

### Reporting summary

Further information on research design is available in the Nature Portfolio Reporting Summary linked to this article.

## Data availability

The NGS data generated in this study have been deposited in the ENA (European Nucleotide Archive) database under accession code PRJEB76066. All other data generated in this study are provided in the Supplementary Information/Source Data file.

## Code availability

No specific code was generated for this study.

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

## Acknowledgements

We thank the staff of the animal facility, the EM facility, and the "genome analytics core facility" of the HZI for technical support. We thank Achim Gronow for his helping hands. M.W. was supported by the federal state Saxony-Anhalt and the European Structural and Investment Funds (ESF, 2014-2020, project number 44 100 32 030 ZS/2016/08/80645). T.S. was supported by the Joint Programming Initiative on Antimicrobial Resistance (project number 01KI1824), the BMBF (DF-AMR2: project number 01KI2131), the German Center for Infection Research (project number 06.826), and the Deutsche Forschungsgemeinschaft (DFG, German Research Foundation - EXC 2155 - project number 390874280). B.F.D. was supported by the Hannover Biomedical Research School (HBRS), the Center for Infection Biology (ZIB), and the Graduate School Scholarship Program (GSSP) from the German Academic Exchange Service (DAAD). The funders did not influence study design, data collection and analysis, or the publishing process.

## Author contributions

M.W. and T.S. designed the experiments and wrote the manuscript with input from co-authors. M.W. and L.O. performed animal experiments, and M.W. conducted most analyses. M.W. and L.E. generated knock-outs for *E. coli*, and L.E. helped with ex vivo and in vitro assays. T.R.L. and U.M. established the 16S rRNA gene sequencing pipeline and supported it with bioinformatical analysis. A.A.B. extracted DNA and conducted library preparation for WGS. M.G. and B.F.D. conducted further phylogenetic analysis and GWAS analysis. E.A. helped with the knockout system establishment. K.A.W., J.S., N.P., S.G., K.S., and D.S. provided *E. coli* human isolates.

## Funding

## Competing interests

T.S., M.W., and L.O. filed a provisional patent for using *E. coli* strains to decolonize MDR Enterobacteriaceae from the gut (EP24182102.4/ PCT/ EP2025/060744). All other authors do not declare any competing interests.
