## [Transparent Peer Review file · Nature Communications]

Suppression of gut colonization by multidrug-resistant *Escherichia coli* clinical isolates through cooperative niche exclusion

Corresponding Author: Professor Till Strowig

Version 0:

Reviewer comments:

Reviewer #1

(Remarks to the Author)

“Gut decolonization of multidrug-resistant *Escherichia coli* clinical isolates via cooperative niche exclusion” is a manuscript that identifies and shows the efficacy of novel probiotic strains against MDR *E. coli* in vitro and in mouse models and delve into mechanisms of niche exclusion via carbohydrate metabolism. Despite the title suggesting a therapeutic application, this paper is largely focused on a preventative approach requiring precolonization/establishment of the probiotic strain as loosely-defined microbiome influences. The investigators also identified differences in carbon utilization (specifically mannose) as a driving force behind commensal strains outcompeting a representative MDR strain. By extension, increasing the metabolic range of the probiotic pre-treatment (by using two probiotic strains) increases the protective range of this preventative approach.

Major concerns:

- 1) The language used in the manuscript generously applies terms like “decolonization” and “clearance,” in some cases these terms are not supported by the data presented in the manuscript. For example, there is one main figure that discusses a decolonization approach (where mice pre-colonized with MDR *E. coli* are given a probiotic strain), while the others show data from a model of probiotic prophylaxis. Consider tempering the language in the manuscript, including the title, to better reflect the data. An example for a title that more accurately reflects the work done would be “Suppression of multi-drug resistant *Escherichia coli* isolates via cooperative niche exclusion.”
- 2) The authors should revise the manuscript to uniformly add descriptions of statistical tests that back up their claims and make any necessary alterations to the manuscript if the statistics do not support the conclusions. Including:
 - a. Line 279-280 and Figure 3D: The reviewer cannot see the “distinct patterns of recovery between control and MR102-colonized mice”. The points for both of the groups seem to overlap for any given time point. Performing a statistical test (like a PERMANOVA) with respect to the condition and time would be useful.
 - b. Line 303-305: Were any statistical tests done on these comparisons?
 - c. Line 408: There are no statistical tests to back up these conclusions
 - d. Figure 6C-G: Were any follow-up multiple comparisons done after the two-way ANOVA to identify which comparisons are driving the significance seen in the ANOVA?

Minor concerns:

The reviewer suggests moving the data from GF and OMM mice (current figure 3A-3C) to the supplement, as these panels disrupt the flow between 16S data and the cfu data from the corresponding mice in figure 2.

Suggest moving the data from GF and OMM mice to supplement to better represent the flow of experiments from Figure 2 to the corresponding 16S data.

Methods: What vendor do the mice come from, do the SPF mice have no enterobacteriaceae?

Line 137: What is the rationale behind using a 1:10 MDR: commensal ratio?

Line 138-139: The use of a cecal content from GF mice to mimic a “heavily disturbed microbiota after antibiotic treatment”

does not seem appropriate, as mice treated with antibiotics are not germ-free. Even after antibiotic treatment there are still microbes present and by extension, possible metabolites that could impact growth. Additionally, how were differences between individual mice cecal content compositions controlled for across all samples/replicates of this experiment? Were cecal contents pooled and the same pool used throughout the work?

Line 143-144: What was the antibiotic/selective plating used to distinguish between E.coli Nissle 1917 and E.coli MDR1? The antibiotics/selective plating used throughout this paper is unclear.

Line 154-162: What phylogroup does MDR1 belong to? Is there a correlation/relationship between its phylogroup and those of the competitive commensals?

Line 172: There is no Figure S1H in the supplement

Line 173-175: State where the data showing this comparison is located.

Line 198-200: How are the commensal strain and MDR1 strains distinguished for CFU counting? What are the respective antibiotics being used if all strains have ampicillin resistance?

Line 203-204 and Figure 2B: Figure 2B is missing legend for both the meaning of the colors and the shapes. In addition, the colors and shapes do not 100% match the legend that is provided in Figure 2C so it is unclear which each line is supposed to represent.

Figure 2B and 2C: Assuming that the legend in 2C is meant for both 2B and 2C, there seems to be a discrepancy between the two with regards to MR158 and RV228. At Day 42, their CFU/g average seems to be at the limit of detection. Wouldn't this suggest that the clearance should be at 100% as there's no detectable MDR1?

Line 204-107 and Figure S2: How do the CFU levels of the commensals change over time? Do they remain consistent throughout or do they also seem to be cleared over time like MDR1? This could suggest that the mouse GI is clearing both strains as its GI recovers after ampicillin treatment has stopped.

Line 227-229 and Figure 2F: Is it expected that MR102 be still detectable throughout the experiment as compared to the prophylactic experiment? This can be strengthened by including the CFU data for MR102 in the prophylactic experiments.

Line 234-235: MR102 seems to be even more effective against MDR2 compared to MDR1, does this track with what you see later about the carbohydrate utilization?

Line 263: What is the strain and phylogroup of the native E.coli strain in OMM19? Did you test this strain's ability to reduce MDR1 CFU in your ex vivo model?

Line 227-280 and Figure 3D: Why is data only up to day 9 shown on the beta diversity plot? Looking at the longitudinal data from Figure 2B and 2C, we don't start seeing those higher levels of clearance (that sets MR102 apart from the other competitive strains) until around Day 14. Additionally, having the data from the final time point (Day 42) would be useful to see how the microbiome recovers over the entire course of the experiment. Another time point that would be useful to include is Day -3 (right after inoculation with the commensal) to show how this affects the microbiome.

Line 281-283 and Figure 3F: The difference between the two is not statistically distinguishable (following the key provided in the figure legend). This figure may also be strengthened with the addition of more later time points. The data are more compelling (and statistics back up) conclusions from Figure 3E in this regard.

Line 293: Why was Day 9 selected for the metagenome sequencing?

Line 295: Figure 3E is referenced here but Figure 3E references alpha diversity changes via 16S, not function via metagenomics.

Line 300-301: If the mice were grouped based on their status at Day 14, why weren't fecal samples from Day 14 also used? Are there expected differences between Day 9 and Day 14?

Line 344-346: Of the 6 unitigs that mapped to phylogroup D, which were associated with carbon utilization?

Line 379-381: When was cellobiose stopped? Unclear from Figure 4C.

Figure 5A: Figure legend has "MHH" written for MDR1 but this abbreviation is used nowhere else in the paper.

Line 410: Of the three sugars identified previously, why was mannose selected?

Line 412-415 and Figure S5C: Why is a 1:10 commensal:MDR ratio used for the ex vivo competition assays but 1:1 is used for the minimal media assays?

Line 434: Why was a 1:10 ratio used? Could you see the same results in a 1:1 ratio?

Line 438-440: How much of a reduction in CFU is counted as a positive hit?

Line 462-365: What does MR102 and MK01 colonization look like in monoculture? A better comparison to show that the two strains do not interfere with each other would be to compare their co-culture levels to their mono-culture levels.

Line 471-473: The in vitro results suggested that MDR3 would not be inhibited by MR102 and MK01. How do you explain the difference between these results?

Line 473-476 and Figure 6E: It seems that MDR3 is just not a good GI colonizer. Can the authors expand on this idea?

Reviewer #2

(Remarks to the Author)

The authors screened human commensal *E. coli* strains to identify those that could promote decolonization of multi-drug resistant *E. coli*. About 10% of strains tested inhibited growth of a model MDR strain in cecal contents prepared from germ free mice. About 50% of the competitive strains were in phylogroups B1 and D. Genome analysis of virulence and antimicrobial genes did not reveal anything interesting. When precolonized, the most inhibitory strains were able to "decolonize" an MDR strain in ampicillin treated mice, leading to clearance in many cases. Experiments in germ free or OMM mice demonstrated lesser inhibition of the MDR strain and no clearance, indicating that decolonization required a more complex microbiota. Carbohydrate utilization analysis indicated the inhibitory strains tended to use more carbon sources than the MDR strain. However, the MDR strain used cellobiose better. Inclusion of cellobiose in the drinking water during competition somewhat lessened inhibition. Knocking out mannose utilization in the inhibitory strain significantly impacted competition, indicating that carbohydrate utilization contributes to decolonization. Lastly, a combination of inhibitory *E. coli* plus *Klebsiella oxytoca* were more (completely) effective in clearing MDR strains, indicating that niche-exclusion by diverse Enterobacteriaceae has the greatest potential for decolonizing MDR *E. coli*.

The results are important because they provide evidence that a diverse healthy microbiota can prevent colonization by and decolonize MDR *E. coli*, quite possibly through a mechanism that involves carbohydrate niche exclusion. The screening of hundreds of strains in vitro to identify and focus on a tractable number of inhibitory strains is innovative. All in all, this is an exciting contribution to microbiome science.

Major comment: The data are solid. The controls performed exactly as expected. The design of the experiments and results are thoroughly convincing.

Reviewer #3

(Remarks to the Author)

In the manuscript from Wende et al., the authors screen a collection of *E. coli* isolated from the gut for their ability to outcompete multidrug-resistant *E. coli* strains (MDR-E). A subset of these commensal isolates (with varying capacities to inhibit an MDR-E strain) were tested in antibiotic-treated mice for their ability to decolonize an MDR-E strain. This decolonization capacity seems to be dependent on the background gut microbiota, since in germ-free mice or mice colonized with Oligo-MM12 or Oligo-MM19 decolonization does not occur. Competition for carbon sources is shown to drive the interaction observed between commensal *E. coli* and MDR-E in vitro. Lastly, combination of commensal *E. coli* with *Klebsiella oxytoca* (previously shown to decolonize *K. pneumoniae* and *Salmonella Typhimurium*) showed a greater effect in decolonization of different MDR species.

The study has the merit to further identify more commensal Enterobacteriaceae species with potential to be used as live biotherapeutics to treat infections with pathogenic Enterobacteriaceae species. This is a relevant area of research, especially in an era of extensive use of antibiotics that disrupt the gut microbiota as well as the rise of antibiotic-resistant bacteria.

However, there are a few important issues that need to be addressed particularly regarding the inconsistency of experimental groups and experimental designs used and proper explanation for taking specific directions, as well as some inconsistency between in vitro and in vivo data that needs to be addressed.

Major

- Throughout the manuscript the authors present several experiments where the used experimental groups are inconsistent, which makes the paper difficult to follow and to make proper comparisons. Moreover, a lot of decisions made throughout the paper lack some proper explanations that are important for the rationale of the experiments. For instance:

- In Figure 1 the authors show that several *E. coli* strains greatly outcompete an MDR strain. The obvious choice to test in vivo would be one of the 2 strains that most outcompete the MDR strain. Why was this not the case?

- Additionally, in Figure 1 the authors assign the different commensal strains into different phylotypes. It remains unclear what (if any) is the relevance of this classification for the competitiveness of the commensal strains. In part, this might be due to the choices made in several experiments. It is described that B1 phylotype contains the greatest % of competitive strains, but since competitive and non-competitive strains of the same phylogroups were not used to test in vivo and in vitro the phylogroup narrative seems unnecessary for the story, since it does not seem to translate into the quality of the strain competitiveness. A proper comparison between non-competitive and competitive strains of the same phylotypes needs to be conducted in mice to reach a conclusion whether this is relevant or not. Also, the authors ended up using mostly a single strain from group D throughout the text.
- The phylogroup analysis appears to lack a more systematic approach, namely the lack of assignment of phylogroups for the MDR strains and consequent analysis; the aforementioned lack of pairing of competitive and non-competitive strains from same phylogroups; and lack of table with all the strains tested with their corresponding FC value (Figure 1b) and phylogroup, so the reader can analyze the data without the subjective 10% threshold the authors use to define competitiveness, which does not seem to follow any clustering in this figure, with several strains with very similar FC values being differently qualified as competitive and non-competitive.
- In Figure 5 the non-competitive strain (LK192) used is not the same as the one used in Figure 6 (MR103). What is the reason for this? Additionally, in Figure 6, besides MR102, it was used another competitive strain (MR134) for which we have no information about their in vivo behavior. Why not use one of the other previously tested competitive strains (MR158, RV228)? Are the different non-competitive strains not behaving similarly in the different assays?
- Some competition assays were performed using a 1:1 initial inoculum of commensal to MDR ratio, while others were performed using 10:1. Again, why was this done differently? This should all be done in the same conditions because it makes comparisons difficult.
- The initial experiment in vitro in Figure 1 is described to be done in microaerophilic conditions. Later the author test aerobic and anaerobic conditions, but not in comparison with microaerophilic conditions. What is the reason for this and what is the relevance of observed higher inhibition capacity of the competitive strain in the presence of oxygen in the context of the gut environment?
- Justify the choice of different AUC thresholds to be considered as growth (e.g. Figure 4b and Figure S6).
- In Figure 2, the authors test several commensals as preventive treatment against MDR1 and choose the best candidate (MR102) for therapeutic treatment against MDR1. They then perform the preventive treatment experiment with MDR2, but did not test the therapeutic treatment, which is a potentially more relevant treatment in the field. The same happened in Figure 6 with the mixed probiotic strains, the therapeutic intervention was not tested.
- The authors conduct a series of in vitro analysis that are used to guide their in vivo experiments. But several discrepancies between in vitro and in vivo experiments are observed, which raises the question if the in vitro assays used are sufficient or adequate to predict in vivo outcomes

- One of the most interesting findings in the manuscript is the importance of the microbiome for the presented phenotypes, which was not sufficiently explored by the authors. What is the difference in the microbiomes between the treatments with all competitive vs non-competitive strains? For example, *Lactobacillus* is found to be associated with a faster decolonization of the MDR species. Is this consistent across microbiomes treated with the different competitive strains? The authors mention that Oligo-MM contains a *Lactobacillus* species and therefore discard the relevance of this species in the phenotype observed when a competitive *E. coli* is present. This is an argument that goes against the whole rationale of the manuscript where the authors find that strains variations at the species level can lead to different protective capacities and it is not described whether the *Lactobacillus* strain in antibiotic-treated mice is the same one that is present in the Oligo-MM. Additionally, it is disregarded the context of these 2 experiments, because the *Lactobacillus* in the antibiotic-treated mice interact with other commensals not present in the Oligo-MM. Previous literature had identified *Lactobacillus* as a species implicated in providing colonization resistance to *K. pneumoniae*, and it is an already established probiotic genus. It would be important to understand if there is a positive interaction between *Lactobacillus* and competitive *E. coli* to outcompete MDR species. Supplementation of the *Lactobacillus* (isolated from the mouse experiment enriched in *Lactobacillus* in the presence of MR102) in combination with MR102 to the antibiotic-treated mouse model or to Oligo-MM would elucidate the role of this member of the microbiome as indicated by the authors' LefSe analysis.

- The authors have recently published another paper showing again the importance of the nutrient competition for the colonization resistance / displacement phenotype among Enterobacteriaceae, which they use as a reference in this manuscript. In that paper, the authors also demonstrate a partial role of a toxin produced by MK01 strain. Is this toxin playing a role in the last experiment, where MK01 is tested, since the wildtype strain is used? And, according to that paper, should the safety of MK01 strain be discussed as a member of a probiotic cocktail, as is tested in this manuscript, since it appears to produce a cytotoxic compound?

Minor

- Data on unitigs (Fig4A) seems irrelevant for the rest the main narrative of the manuscript and lacks a better explanation; should not be in a main figure
- AUC values can be quite subjective and depict completely different growth phenotypes. The growth kinetics should be added as supplementary figures.
- Several statements regarding the microbiome analysis (including Fig3D, G, H and I) lack proper statistical tests.
- Throughout the manuscript figure legends need to be improved to explain all relevant aspects represented (e.g. EcN being the yellow dot in Figure 1, what is MHH and why not use MDR1 like in other figures)
- Material and methods need improvement, namely making sure all numbers of animals used for all experiments are reported, also the calculations for cultures used (in particular in the 10:1 ratio, which appears to be inverted between commensal and MDR strains).

- Present other metrics of diversity, like the Shannon index, for PBS vs MR102 in Figure 3
- Show pre-colonization day for the alpha diversity in Figure 3E
- Show PBS microbiome in pre-colonization time-point (also refer to what day it refers to) in Figure S3
- 1% cellobiose was supplemented to the mice and not 10% as is depicted in the figure (10g/l)
- Explain if OMM mice were from colonies of mice raised with OMM as microbiota or if these are ex-germ-free mice colonized with OMM communities for the experiments
- There is no E panel in Supp figure 2, but there a panel E is described in the legend.
- What is the number of mice used for Fig5?

Reviewer #4

(Remarks to the Author)

Version 1:

Reviewer comments:

Reviewer #1

(Remarks to the Author)

The authors thoroughly address my concerns from the initial submission and I am excited to see this paper published.

Reviewer #3

(Remarks to the Author)

I appreciate the replies from the authors and the additional experiments performed that filled some gaps. I believe that most questions that were raised were sufficiently addressed. Nevertheless, some responses still require some changes in the text. Furthermore, some experiments and/or analyses that were performed and sent to reviewers but not added to the manuscript should also be part of the final manuscript. Please find the detailed replies to the author's responses in the attachment.

Reviewer #4

(Remarks to the Author)

Reviewer comments: Point-by-point response

Reviewer #1 (Remarks to the Author):

“Gut decolonization of multidrug-resistant *Escherichia coli* clinical isolates via cooperative niche exclusion” is a manuscript that identifies and shows the efficacy of novel probiotic strains against MDR *E. coli* in vitro and in mouse models and delve into mechanisms of niche exclusion via carbohydrate metabolism. Despite the title suggesting a therapeutic application, this paper is largely focused on a preventative approach requiring precolonization / establishment of the probiotic strain as loosely-defined microbiome influences. The investigators also identified differences in carbon utilization (specifically mannose) as a driving force behind commensal strains outcompeting a representative MDR strain. By extension, increasing the metabolic range of the probiotic pre-treatment (by using two probiotic strains) increases the protective range of this preventative approach.

Author's response: We appreciate the reviewer's constructive criticism and have incorporated additional experiments and analyses to support our findings.

Major concerns:

1) The language used in the manuscript generously applies terms like “decolonization” and “clearance,” in some cases these terms are not supported by the data presented in the manuscript. For example, there is one main figure that discusses a decolonization approach (where mice pre-colonized with MDR *E. coli* are given a probiotic strain), while the others show data from a model of probiotic prophylaxis. Consider tempering the language in the manuscript, including the title, to better reflect the data. An example for a title that more accurately reflects the work done would be “Suppression of multi-drug resistant *Escherichia coli* isolates via cooperative niche exclusion.”

*Author's response: The reviewer is correct that the majority of experimental work was conducted in a prophylactic context, but key findings were also confirmed in a therapeutic setting, including in additional experiments displayed in the revised Figure S2 E, F and S7 F-K. Importantly, our in vivo data show that in the absence of the commensal *E. coli*, the MDR strains are successfully integrated into the microbiota and do not clear spontaneously from the gut until 6 weeks of colonization. We, therefore, argue that terms such as decolonization and clearance can still be used throughout the manuscript to describe the removal of the MDR-E strains from the gut. Nevertheless, to properly reflect the main findings of the manuscript, we have, as suggested, renamed the manuscript “Suppression of multidrug-resistant *Escherichia coli* clinical isolates via cooperative niche exclusion”.*

2) The authors should revise the manuscript to uniformly add descriptions of statistical tests that back up their claims and make any necessary alterations to the manuscript if the statistics do not support the conclusions. Including:

Author's response: We have reviewed the manuscript and included throughout the manuscript the results of the respective tests (if not already listed, see revised figure 3A, B, 3C, D, E, F, G, 5C).

a. Line 279-280 and Figure 3D: The reviewer cannot see the “distinct patterns of recovery between control and MR102-colonized mice”. The points for both of the groups seem to overlap for any given time point. Performing a statistical test (like a PERMANOVA) with respect to the condition and time would be useful.

Author's response: *In response to this and another reviewer's comment (Reviewer 3), we have included additional commensal strains in the analysis displayed in Figure 3D (revised Figure 3A, C, D). Specifically, we now focus on day 6 (the first timepoint after antibiotic treatment was stopped). We observed significantly different results for the various groups based on our analysis using Permanova (β -diversity) and the Mann-Whitney test (α -diversity). The results are also mentioned in line 258-268.*

b. Line 303-305: Were any statistical tests done on these comparisons?

Author's response: *We now included statistical tests (revised Fig 3F Permanova and Fig 3G Mann-Whitney test) for the data.*

c. Line 408: There are no statistical tests to back up these conclusions

Author's response: *We now included an additional biological replicate and included statistical tests (Mann-Whitney test, see revised Fig. 5A).*

d. Figure 6C-G: Were any follow-up multiple comparisons done after the two-way ANOVA to identify which comparisons are driving the significance seen in the ANOVA?

Author's response: *Yes, the multiple comparisons are included in the source data file. You can find the results of the multiple comparisons for MDR3, as an example, below. We could observe significant differences at day 1 comparing PBS vs. mix-treated groups, MK01 vs. MR102-treated groups, and MR102 vs. mix-treated groups. For day 3 PBS vs. MK01, PBS vs. mix, MK01 vs. mix, and MR102 vs. mix. Especially the significant differences comparing mix to MK01 or MR102-treated groups at day 3 indicates accelerated reduction in the mix-treated group, compared to single-strain treated groups.*

E. coli MDR3						
Tukey's multiple comparisons test	Mean Diff,	95,00% CI of	Below thresh	Summary	Adjusted P Value	
Day 1						
PBS vs. K. oxytoca MK01	8670500000	-221198714 to	No	ns	0,0556	
PBS vs. E. coli MR102	6950375000	-1939959724 to	No	ns	0,1307	
PBS vs. mix	9116322665	223427567 to	Yes	*	0,0449	
K. oxytoca MK01 vs. E. coli MR102	-1720125000	-3203360031 to	Yes	*	0,0234	
K. oxytoca MK01 vs. mix	445822665	-161148018 to	No	ns	0,1581	
E. coli MR102 vs. mix	2165947665	702519388 to	Yes	**	0,0073	
Day 3						
PBS vs. K. oxytoca MK01	1,3497E+10	879654169 to	Yes	*	0,0371	
PBS vs. E. coli MR102	9451125000	-3330072158 to	No	ns	0,1685	
PBS vs. mix	1,4733E+10	2113395673 to	Yes	*	0,0248	
K. oxytoca MK01 vs. E. coli MR102	-4045375000	-9213672966 to	No	ns	0,1315	
K. oxytoca MK01 vs. mix	1236337001	120107230 to	Yes	*	0,0317	
E. coli MR102 vs. mix	5281712001	114416601 to	Yes	*	0,0455	
Day 6						
PBS vs. K. oxytoca MK01	1531578409	-474759133 to	No	ns	0,1394	
PBS vs. E. coli MR102	1524254114	-482063763 to	No	ns	0,1416	
PBS vs. mix	1550498323	-455879289 to	No	ns	0,1338	
K. oxytoca MK01 vs. E. coli MR102	-7324295	-76839220 to	No	ns	0,9894	
K. oxytoca MK01 vs. mix	18919914	-30862389 to	No	ns	0,6139	
E. coli MR102 vs. mix	26244209	-34790783 to	No	ns	0,5248	
Day 9						
PBS vs. K. oxytoca MK01	48467945	-71208288 to	No	ns	0,5878	
PBS vs. E. coli MR102	57769850	-61772105 to	No	ns	0,4363	
PBS vs. mix	57873830	-61668132 to	No	ns	0,4349	
K. oxytoca MK01 vs. E. coli MR102	9301905	-21218078 to	No	ns	0,7498	
K. oxytoca MK01 vs. mix	9405885	-21114125 to	No	ns	0,7439	
E. coli MR102 vs. mix	103980	-55108 to 263	No	ns	0,2227	

Minor concerns:

The reviewer suggests moving the data from GF and OMM mice (current figure 3A-3C) to the supplement, as these panels disrupt the flow between 16S data and the cfu data from the corresponding mice in figure 2.

Suggest moving the data from GF and OMM mice to supplement to better represent the flow of experiments from Figure 2 to the corresponding 16S data.

Author's response: *Incorporating the comments from this and other reviewers, we now reordered the data presentation and interpretation in Figure 3, including moving the data from GF and OMM mice to the revised Figure S3.*

Methods: What vendor do the mice come from, do the SPF mice have no enterobacteriaceae?

Author's response: *All SPF mice were bred internally at the animal facility of the Helmholtz Centre for Infection Research. They were originally rederived through embryo transfer more than ten years ago (Stehr et al., Laboratory Animals, 2009) and have been used routinely over the last few years as a microbiota model community to study colonization resistance (e.g., Thiemann et al., Cell Host Microbe, 2017; Osbelt et al., Cell Host Microbe, 2021; Almasi et al., Nat Comm, 2025). These mice do not harbor any Enterobacteriaceae, as documented through*

both NGS and cultivation-based methods in previous studies. Moreover, in this study, the absence of Enterobacteriaceae was routinely confirmed before each experiment by plating fecal samples of untreated mice on MacConkey agar.

Line 137: What is the rationale behind using a 1:10 MDR: commensal ratio?

Author's response: *To determine the ratio for the ex vivo assay in cecum content of animals different ratios of MDR and commensal E. coli were tested in a titration experiment. A 1:1 (MDR1: MR102) to 1:50 (MDR1: MR102) ratio was tested (revised Figure S1C). As we consider a reduction of CFUs of more than 10-fold a robust reduction of CFU/ ml and competitive effect, we selected the 1:10 ratio for all ex vivo experiments in isolated cecum content.*

Line 138-139: The use of a cecal content from GF mice to mimic a “heavily disturbed microbiota after antibiotic treatment” does not seem appropriate, as mice treated with antibiotics are not germ-free. Even after antibiotic treatment there are still microbes present and by extension, possible metabolites that could impact growth. Additionally, how were differences between individual mice cecal content compositions controlled for across all samples/replicates of this experiment? Were cecal contents pooled and the same pool used throughout the work?

Author's response: *We agree that GF cecum content is not identical to the content of antibiotic-treated mice, however, it is an approximation. Specifically, after ampicillin treatment of our SPF mice, microbial loads are dramatically reduced (>100-fold) as indicated by reduced DNA yield from feces. Of note, a recent study by the groups of W.D. Hardt and U. Sauer demonstrated that monosaccharide concentrations are more similar between GF and streptomycin-treated mice compared to OMM12 or untreated SPF mice. Hence, we are convinced that the use of GF cecum content enables us to mimic competition in the gut environment in an ex vivo setting better than the classical microbiological media. Throughout the experiments, we used pools of cecum content from different mice, but since experiments were performed over a period of time, different pools were used. To control for variations throughout the work, we kept the strain MR102 as a control strain in different assays (reviewer figure 1). Indeed, we could observe minor differences between assays, as we observed in in vivo experiments and as we would expect while working with animals or animal derived samples. We have rephrased the sentence to the following, see line 140: “germ-free (GF) mice imitating the nutritional landscape of the gut environment without the interference of other bacteria”*

MR102 vs. MDR1 in different pools of cecum content

Reviewer Figure 1: FCgrowth of *E. coli* MDR1 in co-cultivation with MR102 in seven different pools of cecum content.

Line 143-144: What was the antibiotic/selective plating used to distinguish between *E. coli* Nissle 1917 and *E. coli* MDR1? The antibiotics/selective plating used throughout this paper is unclear.

Author's response: We did not enumerate the commensal *E. coli* strains in the ex vivo assay (Fig. 1B). The MDR *E. coli* was identified by plating on selective agar plates, as MDR *E. coli* strains inherently possess various resistance genes. We could differentiate all MDR *E. coli* strains from commensal *E. coli* strains by plating on LB agar plates supplemented with ciprofloxacin.

Line 154-162: What phylogroup does MDR1 belong to? Is there a correlation/relationship between its phylogroup and those of the competitive commensals?

Author's response: The *E. coli* MDR1 belongs to Phylogroup A and competitive strains are enriched in phylogroups B1 and D. This is now reported in lines 142, 198-200, and 498. To test whether a correlation between phylogroup of MDR and competitive commensals exist, additional extensive studies would be required.

Line 172: There is no Figure S1H in the supplement

Author's response: We corrected Figure S1 and included the missing "H". Figure S1H shows the ARGs of commensal and BSI *E. coli* isolates.

Line 173-175: State where the data showing this comparison is located.

Author's response: *The data is shown in Figure S1E (VAGs) and S1F (ARGs). This is corrected in the manuscript, see line 164-176.*

Line 198-200: How are the commensal strain and MDR1 strains distinguished for CFU counting? What are the respective antibiotics being used if all strains have ampicillin resistance?

Author's response: *We did not enumerate the commensal E. coli strains in the ex vivo assay (Fig. 1B). The MDR E. coli was distinguished by plating on selective agar plates. MDR E. coli strains inherently possess various resistance genes, but we could not differentiate all MDR E. coli strains from commensal E. coli strains by plating on LB agar plates supplemented with ciprofloxacin. Later, we were able to discriminate at least MR102 from MDR1 by plating on MacConkey agar supplemented with D-maltose, since MDR1 cannot utilize D-maltose as a sole carbon source. This distinction was not routinely possible in experiments with various commensal strains as they partly lacked metabolic features, allowing simple discrimination.*

Line 203-204 and Figure 2B: Figure 2B is missing legend for both the meaning of the colors and the shapes. In addition, the colors and shapes do not 100% match the legend that is provided in Figure 2C so it is unclear which each line is supposed to represent.

Author's response: *Thanks for catching this oversight. We expanded the legend for Figure 2B, and the legends from Figures 2B and C now fully match the represented data.*

Figure 2B and 2C: Assuming that the legend in 2C is meant for both 2B and 2C, there seems to be a discrepancy between the two with regards to MR158 and RV228. At Day 42, their CFU/g average seems to be at the limit of detection. Wouldn't this suggest that the clearance should be at 100% as there's no detectable MDR1?

Author's response: *We made a mistake in defining the detection limit in the graph, which resulted in improper display and misinterpretation. This has now been corrected, and both strains now display low colonization levels of MDR1.*

Line 204-107 and Figure S2: How do the CFU levels of the commensals change over time? Do they remain consistent throughout or do they also seem to be cleared over time like MDR1? This could suggest that the mouse GI is clearing both strains as its GI recovers after ampicillin treatment has stopped.

Author's response: *We totally agree that the colonization levels of the commensal isolates over time is a very interesting information, however, at the time we performed these experiments, we were not able to discriminate the commensal strain from the MDR1 isolate by plating. Later, we were able to discriminate at least MR102 from MDR1 by plating on MacConkey agar supplemented with D-maltose, since MDR1 cannot utilize D-maltose as a sole carbon source (therapeutic experiment, Figure 2 D-F). At this later timepoint we are not able to retrospectively generate this data.*

Line 227-229 and Figure 2F: Is it expected that MR102 be still detectable throughout the experiment as compared to the prophylactic experiment? This can be strengthened by including the CFU data for MR102 in the prophylactic experiments.

Author's response: As described above, we totally agree that this would be a piece of valuable information, but we were not yet able to discriminate both strains when we performed the prophylactic experiments.

Line 234-235: MR102 seems to be even more effective against MDR2 compared to MDR1, does this track with what you see later about the carbohydrate utilization?

Author's response: We included the *E. coli* MDR2 strain in the Biolog assay and compared its carbohydrate utilization pattern to the other commensal and MDR strains (revised Figure S6). In line with the *in vivo* results of faster decolonization of MDR2 by MR102 than MDR1, we identified that MDR1 has a broader carbohydrate utilization capacity compared to MDR2, yet, MR102 can utilize the most carbohydrates. This is also described in the text in line 445

Line 263: What is the strain and phylogroup of the native *E. coli* strain in OMM19? Did you test this strain's ability to reduce MDR1 CFU in your *ex vivo* model?

Author's response: We tested the *E. coli* Mt1B1 (phylogroup B2) strain in our *ex vivo* model and could demonstrate a very slight reduction of CFUs compared to the control group, but not as strong as we could observe for the MR102 strain (see Reviewer Figure 2).

Ex vivo assay *E. coli* Mt1B1 from OMM19

Reviewer Figure 2: CFU/ ml of coli MDR1 after co-cultivation with commensal strains in isolated cecal contents of GF animals in a 1:10 ratio for 24h of microaerophilic cultivation. CFUs of MDR *E. coli* were quantified by plating on selective agar plates.

Line 227-280 and Figure 3D: Why is data only up to day 9 shown on the beta diversity plot? Looking at the longitudinal data from Figure 2B and 2C, we don't start seeing those higher levels of clearance (that sets MR102 apart from the other competitive strains) until around Day 14. Additionally, having the data from the final time point (Day 42) would be useful to see how the microbiome recovers over the entire course of the experiment. Another time point that

would be useful to include is Day -3 (right after inoculation with the commensal) to show how this affects the microbiome.

Author's response: *We only show the recovery until day nine, as these are the most important time points immediately after switching the mice back to normal water. Additionally, day nine is the first day when clearance occurs. We included day -3 in the relative abundance plot (revised Figure S3G), which shows that around 90 % of detected bacterial families belong to Enterobacteriaceae. We needed to exclude the PBS-treated group from the analysis because we did not obtain sufficient reads (<700 reads/sample) for the analysis. Since we observed almost complete recovery already at day 28, we decided not to include day 42 in the microbiome analysis.*

Line 281-283 and Figure 3F: The difference between the two is not statistically distinguishable (following the key provided in the figure legend). This figure may also be strengthened with the addition of more later time points. The data are more compelling (and statistics back up) conclusions from Figure 3E in this regard.

Author's response: *We included now also days 14 to 28 in the analysis (revised Figure 3E). wUnifrac distances are not significantly decreased but show a trend to be smaller. The text has been adjusted accordingly.*

Line 293: Why was Day 9 selected for the metagenome sequencing?

Author's response: *We selected day nine, which marks the first day when clearance occurs. Furthermore, it was observed that microbiome recovery influences the protective effect, and microbiome recovery is still ongoing on day nine .*

Line 295: Figure 3E is referenced here but Figure 3E references alpha diversity changes via 16S, not function via metagenomics.

Author's response: *Thank you for bringing this to our attention. We referred to the incorrect panel; it should have been Figure S3K*

Line 300-301: If the mice were grouped based on their status at Day 14, why weren't fecal samples from Day 14 also used? Are there expected differences between Day 9 and Day 14?

Author's response: *We grouped mice based on their status on day 14 and took the feces from day 9, since day 9 is the first day were clearance is happening. As we could observe, mice that showed clearance at day 14 already had low colonization levels on day 9.*

Line 344-346: Of the 6 unitigs that mapped to phylogroup D, which were associated with carbon utilization?

Author's response: *The unitigs to the gene fnr (fumarate and nitrate reduction regulator) are directly involved in carbon utilization. Fnr regulates anaerobic respiration and influences metabolic pathways, including carbon utilization under low oxygen conditions.*

Line 379-381: When was cellobiose stopped? Unclear from Figure 4C.

Author's response: *Cellobiose treatment was stopped at day 14.*

Figure 5A: Figure legend has "MHH" written for MDR1 but this abbreviation is used nowhere else in the paper.

Author's response: *This was a typo and has been corrected.*

Line 410: Of the three sugars identified previously, why was mannose selected?

Author's response: *D-Mannose was selected, as an example as a monosaccharide, based on the strong reduction up to 10,000-fold.*

Line 412-415 and Figure S5C: Why is a 1:10 commensal:MDR ratio used for the ex vivo competition assays but 1:1 is used for the minimal media assays?

Author's response: *We used a 1:1 ratio for the minimal media assay, since we wanted to analyze the competition for a sole carbon source, where we can already see different outcomes when incubating in a 1:1 ratio.*

Line 434: Why was a 1:10 ratio used? Could you see the same results in a 1:1 ratio?

Author's response: *To determine the ratio for the ex vivo assay in cecum content of animals different ratios of MDR and commensal E. coli were tested in a titration experiment. A 1:1 (MDR1: MR102) to 1:50 (MDR1: MR102) ratio was tested (revised Figure S1C). . As we consider a reduction of CFUs of more than 10-fold a robust reduction of CFU/ ml and competitive effect, we selected the 1:10 ratio for all ex vivo experiments in isolated cecum content.*

Line 438-440: How much of a reduction in CFU is counted as a positive hit?

Author's response: *As the legend states, a >10-fold reduction is counted as intermediate competition and >100-fold as competition.*

Line 462-365: What does MR102 and MK01 colonization look like in monoculture? A better comparison to show that the two strains do not interfere with each other would be to compare their co-culture levels to their mono-culture levels.

Author's response: *We compared the co-colonization levels to those of the mono-colonization (n=3-5 mice per group) and presented the results in revised Figure 6C. We did not observe significant differences in colonization levels, except for minor variations at single time points.*

Line 471-473: The in vitro results suggested that MDR3 would not be inhibited by MR102 and MK01. How do you explain the difference between these results?

Author's response: *At this point we can predict trends in competition using the ex vivo assay, but apparently they are imperfect. We aim in future studies to better model in vitro the different nutrient environments that these strains encounter in vivo. Moreover, the interaction of the MK01 and MR102 strains with the residual microbiota may have been critical for the successful inhibition, which leads to displacement from the gut.*

Line 473-476 and Figure 6E: It seems that MDR3 is just not a good GI colonizer. Can the authors expand on this idea?

Author's response: *We can observe slightly different levels of gut colonization based on different strains. MDR3 can colonize up to 10^{10} CFUs at the beginning of the experiment and up to 10^5 CFUs until the end. We still think that this is a sufficient gut colonization to model decolonization and we included this in the discussion see line 556-558.*

Reviewer #2 (Remarks to the Author):

The authors screened human commensal *E. coli* strains to identify those that could promote decolonization of multi-drug resistant *E. coli*. About 10% of strains tested inhibited growth of a model MDR strain in cecal contents prepared from germ free mice. About 50% of the competitive strains were in phylogroups B1 and D. Genome analysis of virulence and antimicrobial genes did not reveal anything interesting. When precolonized, the most inhibitory strains were able to “decolonize” an MDR strain in ampicillin treated mice, leading to clearance in many cases. Experiments in germ free or OMM mice demonstrated lesser inhibition of the MDR strain and no clearance, indicating that decolonization required a more complex microbiota. Carbohydrate utilization analysis indicated the inhibitory strains tended to use more carbon sources than the MDR strain. However, the MDR strain used cellobiose better. Inclusion of cellobiose in the drinking water during competition somewhat lessened inhibition. Knocking out mannose utilization in the inhibitory strain significantly impacted competition, indicating that carbohydrate utilization contributes to decolonization. Lastly, a combination of inhibitory *E. coli* plus *Klebsiella oxytoca* were more (completely) effective in clearing MDR strains, indicating that niche-exclusion by diverse Enterobacteriaceae has the greatest potential for decolonizing MDR *E. coli*.

The results are important because they provide evidence that a diverse healthy microbiota can prevent colonization by and decolonize MDR *E. coli*, quite possibly through a mechanism that involves carbohydrate niche exclusion. The screening of hundreds of strains in vitro to identify and focus on a tractable number of inhibitory strains is innovative. All in all, this is an exciting contribution to microbiome science.

Major comment: The data are solid. The controls performed exactly as expected. The design of the experiments and results are thoroughly convincing.

Author's response: We appreciate the reviewer's positive remarks.

Reviewer #3 (Remarks to the Author):

In the manuscript from Wende et al., the authors screen a collection of *E. coli* isolated from the gut for their ability to outcompete multidrug-resistant *E. coli* strains (MDR-E). A subset of these commensal isolates (with varying capacities to inhibit an MDR-E strain) were tested in antibiotic-treated mice for their ability to decolonize an MDR-E strain. This decolonization capacity seems to be dependent on the background gut microbiota, since in germ-free mice or mice colonized with Oligo-MM12 or Oligo-MM19 decolonization does not occur. Competition for carbon sources is shown to drive the interaction observed between commensal *E. coli* and MDR-E in vitro. Lastly, combination of commensal *E. coli* with *Klebsiella oxytoca* (previously shown to decolonize *K. pneumoniae* and *Salmonella Typhimurium*) showed a greater effect in decolonization of different MDR species.

The study has the merit to further identify more commensal Enterobacteriaceae species with potential to be used as live biotherapeutics to treat infections with pathogenic Enterobacteriaceae species. This is a relevant area of research, especially in an era of extensive use of antibiotics that disrupt the gut microbiota as well as the rise of antibiotic-resistant bacteria.

However, there are a few important issues that need to be addressed particularly regarding the inconsistency of experimental groups and experimental designs used and proper explanation for taking specific directions, as well as some inconsistency between in vitro and in vivo data that needs to be addressed.

Author's response: We appreciate the reviewer's positive remarks recognizing the contribution of our work to promoting the removal of MDR E. coli from the gut. Using additional experiments and analyses, we have addressed the issue identified by her/him, as detailed in the specific responses.

Major

- Throughout the manuscript the authors present several experiments where the used experimental groups are inconsistent, which makes the paper difficult to follow and to make proper comparisons. Moreover, a lot of decisions made throughout the paper lack some proper explanations that are important for the rationale of the experiments. For instance:

• In Figure 1 the authors show that several *E. coli* strains greatly outcompete an MDR strain. The obvious choice to test in vivo would be one of the 2 strains that most outcompete the MDR strain. Why was this not the case?

*Author's response: We did not clearly communicate why certain strains were tested and others were not, but we have now clarified this. We selected the strains based on their natural ampicillin resistance, which is necessary for colonization in ampicillin-treated SPF mice. We listed all antibiotic resistance genes in Table S5; from the 15 most competitive strains, four encoded a β -lactamase (*bla*TEM-1). Based on this, we chose three of these strains for in vivo evaluation of the competitive phenotype (see Reviewer Table 1). We have now included an explanation in line 191-193.*

Reviewer Table 1: 15 most competitive strains and ARGs for β -lactamase.

lowest 10%	ID	fold change	Beta-lactamase (not ESBL or carbapenemase)
1	MR134	0,00430872	
2	LK281	0,00496732	
3	LK046	0,00736913	
4	LK023	0,0074094	
5	LK003	0,00775839	
6	LK276	0,00825708	
7	MR102	0,00857718	blaTEM-1
8	MR043	0,00939597	blaTEM-1
9	LK010	0,00971812	
10	LK074	0,01868456	
11	LK287	0,01893246	
12	MK141	0,01966443	
13	RV228	0,01966527	blaTEM-1
14	MR158	0,01967785	blaTEM-1
15	RV154	0,01980474	

- Additionally, in Figure 1 the authors assign the different commensal strains into different phylotypes. It remains unclear what (if any) is the relevance of this classification for the competitiveness of the commensal strains. In part, this might be due to the choices made in several experiments. It is described that B1 phylotype contains the greatest % of competitive strains, but since competitive and non-competitive strains of the same phylogroups were not used to test in vivo and in vitro the phylogroup narrative seems unnecessary for the story, since it does not seem to translate into the quality of the strain competitiveness. A proper comparison between non-competitive and competitive strains of the same phylotypes needs to be conducted in mice to reach a conclusion whether this is relevant or not. Also, the authors ended up using mostly a single strain from group D throughout the text.

Author's response: *The reviewer poses an excellent question that would require a tremendous amount of work if systematically tested. Based on the enrichment of strains from phylogroups B1 and D within the competitive group, the strains used in the in vivo model as protective strains included phylogroups D (MR102, RV228) and B1 (MR158). Intermediate strains are represented by phylogroups A (MK192), B1 (MR103), and B2 (LK091). The non-protective strain belongs to phylogroup B2. To avoid overinterpreting the link between phylogroups and protectiveness, we have added additional statements in the discussion section; see lines 498-500. Systemic testing for each phylogroup- and ideally even STs- using multiple strains per phylogroup/ST, including both protective and non-protective strains in vivo, is clearly beyond the scope of the current study.*

- The phylogroup analysis appears to lack a more systematic approach, namely the lack of assignment of phylogroups for the MDR strains and consequent analysis; the aforementioned lack of pairing of competitive and non-competitive strains from same phylogroups; and lack of table with all the strains tested with their corresponding FC value (Figure 1b) and phylogroup, so the reader can analyze the data without the subjective 10% threshold the authors use to define competitiveness, which does not seem to follow any clustering in this figure, with several

strains with very similar FC values being differently qualified as competitive and non-competitive.

Author's response: *The reviewer is correct in stating that we didn't provide the phylogroups of the MDR strains, which we have now included in both the text and Table S2. However, the FC values, ST, and phylogroup for all strains were already listed in Table S4. We acknowledge that the sheet in Table S3 was named "metadata", which was not as informative as it could be. To avoid future confusion, we have now rearranged all previous sheets of tables into separate tables to facilitate the finding of relevant information. In combination with the genomic information, which is publicly available, the reviewer and other researchers can explore alternative clustering approaches.*

- In Figure 5 the non-competitive strain (LK192) used is not the same as the one used in Figure 6 (MR103). What is the reason for this? Additionally, in Figure 6, besides MR102, it was used another competitive strain (MR134) for which we have no information about their in vivo behavior. Why not use one of the other previously tested competitive strains (MR158, RV228)? Are the different non-competitive strains not behaving similarly in the different assays?

Author's response: *The response to the reviewer's comment is based in part again on the ampicillin resistance/susceptibility of strains, e.g., the strain MR134 lacks an ampicillin resistance and could thus not be tested in the in vivo model. However, we thought it would be informative to screen it in the in vitro assay. To enrich the information of the data shown in Figure 6A we have performed additional experiments for the strain LK192 testing its competitiveness alone and in combination with the MK01 strain. In line with its in vivo phenotype, little competitiveness is detected against the broader set of tested MDR E. coli strains. This information is now included in the revised Figure 6A and line 428f. We are currently considering whether a larger and more systemic screening of combinations of strains with complementary properties both in vitro and in vivo is feasible.*

- Some competition assays were performed using a 1:1 initial inoculum of commensal to MDR ratio, while others were performed using 10:1. Again, why was this done differently? This should all be done in the same conditions because it makes comparisons difficult.

Author's response: *To determine the ratio for the ex vivo assay in cecum content different ratios of MDR and commensal E. coli were tested in a titration experiment. A 1:1 (MDR1: MR102) to 1:50 (MDR1: MR102) ratio was tested (revised Figure S1C). As we consider a reduction of CFUs of more than 10-fold a robust reduction of CFU/ml and competitive effect, we selected the 1:10 ratio for all ex vivo experiments in isolated cecum content. We used a 1:1 ratio for the minimal media assay, since we wanted to analyze the competition for a sole carbon source, where we can already detect different outcomes when incubating in a 1:1 ratio.*

- The initial experiment in vitro in Figure 1 is described to be done in microaerophilic conditions. Later the author test aerobic and anaerobic conditions, but not in comparison with microaerophilic conditions. What is the reason for this and what is the relevance of observed higher inhibition capacity of the competitive strain in the presence of oxygen in the context of the gut environment?

Author's response: *All experiments under anaerobic conditions were performed in plastic boxes with an Anaeropack. This setup allows to modulate anaerobic condition without using an anaerobic chamber. The description was harmonized in the results section and described in detail in the methods section.*

- Justify the choice of different AUC thresholds to be considered as growth (e.g. Figure 4b and Figure S6).

Author's response: *Biolog and growth curve assays cannot be directly compared. Growth curves were performed for 72h and with 5g/L sugar, Biolog assays for 24h, and an sugar concentration, which is not disclosed by the producer.*

- In Figure 2, the authors test several commensals as preventive treatment against MDR1 and choose the best candidate (MR102) for therapeutic treatment against MDR1. They then perform the preventive treatment experiment with MDR2, but did not test the therapeutic treatment, which is a potentially more relevant treatment in the field. The same happened in Figure 6 with the mixed probiotic strains, the therapeutic intervention was not tested.

Author's response: *We agree that the therapeutic treatment may in the future be more relevant than the prophylactic treatment. Hence, we have performed additional in vivo experiments for selected strains also in the therapeutic approach.*

We can specifically demonstrate that E. coli MDR2 is decolonized (100% clearance) after treatment with E. coli MR102. This data is included in Figure S2E and F and is described in line 240f.

Additionally, we conducted further experiments to determine whether the E. coli MDR1, MDR2, and K. pneumoniae strains were eliminated by the combination of E. coli MR102 and K. oxytoca MK01 in a therapeutic treatment setup. We indeed observed that the mixture could reduce the colonization level by 10 to 100 times compared to the PBS-treated group. Furthermore, we noted a 75% clearance rates for E. coli MDR1 and K. pneumoniae, and an 80% clearance rate for E. coli MDR2. This data is presented in revised Figure S7F-K and discussed in line 477.

- The authors conduct a series of in vitro analysis that are used to guide their in vivo experiments. But several discrepancies between in vitro and in vivo experiments are observed, which raises the question if the in vitro assays used are sufficient or adequate to predict in vivo outcomes

Author's response: *Before responding in detail below, we would like to reiterate that the ex vivo assay enables an initial screening of strains regarding their potential to decolonize MDR E. coli. If a strain fails in vitro and is not pursued in vivo, this would not be as detrimental to the development pipeline as if a strain performs well in vitro but then fails in vivo. Notably, as shown in Figures 1 and 2, the assay can predict the in vivo phenotype reasonably well in most cases. However, the assay does not reflect the in vivo phenotype for all tested strains and combinations 100 % of the time, since the environmental conditions in vivo are much more complex (see also below). Overall, we are convinced that the ex vivo assay offers a valuable opportunity to conduct an initial screening of potential probiotic candidates that still require in vivo validation. This is also discussed further in the discussion, see lines 507f.*

- One of the most interesting findings in the manuscript is the importance of the microbiome for the presented phenotypes, which was not sufficiently explored by the authors. What is the difference in the microbiomes between the treatments with all competitive vs non- competitive strains?

Author's response: *Indeed, the contribution of the microbiota to the decolonization of MDR E. coli is relevant, but also highly complex. To provide more insight into microbiota changes in the in vivo model, we performed additional microbiota analysis sequencing on 155 more samples for the strains MR158 and MR103. In general, we observed a strain-dependent*

recovery of the microbiome. Mice treated with MR102 clustered closer to untreated samples on day 6. Furthermore, mice treated with MR102 exhibited significantly higher alpha diversity compared to groups treated with PBS, MR158, MR103, MK192, or LK192. These data are presented in revised Figure 3/S3 and discussed in lines 258-269.

For example, Lactobacillus is found to be associated with a faster decolonization of the MDR species. Is this consistent across microbiomes treated with the different competitive strains?

The authors mention that Oligo-MM contains a Lactobacillus species and therefore discard the relevance of this species in the phenotype observed when a competitive E. coli is present. This is an argument that goes against the whole rationale of the manuscript where the authors find that strains variations at the species level can lead to different protective capacities and it is not described whether the Lactobacillus strain in antibiotic-treated mice is the same one that is present in the Oligo-MM. Additionally, it is disregarded the context of these 2 experiments, because the Lactobacillus in the antibiotic-treated mice interact with other commensals not present in the Oligo-MM. Previous literature had identified Lactobacillus as a species implicated in providing colonization resistance to K. pneumoniae, and it is an already established probiotic genus. It would be important to understand if there is a positive interaction between Lactobacillus and competitive E. coli to outcompete MDR species. Supplementation of the Lactobacillus (isolated from the mouse experiment enriched in Lactobacillus in the presence of MR102) in combination with MR102 to the antibiotic-treated mouse model or to Oligo-MM would elucidate the role of this member of the microbiome as indicated by the authors' LefSe analysis.

Author's response: *[editorial note: reply redacted]*

- The authors have recently published another paper showing again the importance of the nutrient competition for the colonization resistance / displacement phenotype among Enterobacteriaceae, which they use as a reference in this manuscript. In that paper, the authors also demonstrate a partial role of a toxin produced by MK01 strain. Is this toxin playing a role in the last experiment, where MK01 is tested, since the wildtype strain is used? And, according to that paper, should the safety of MK01 strain be discussed as a member of a probiotic cocktail, as is tested in this manuscript, since it appears to produce a cytotoxic compound?

Author's response: *Once again, the reviewer raises another interesting question regarding the potential contribution of the toxin to MDR E. coli inhibition. To test the potential role of tilimycin-production on the effect of the combination of E. coli MR102 and K. oxytoca MK01, we conducted an additional mouse experiment. Specifically, we compared MDR1 colonization levels in the prophylactic model by comparing the inhibitory effect of MR102 together with either the K. oxytoca MK01 WT or $\Delta npsA$ strain. Strikingly, no differences were observed between the groups receiving either the WT or $\Delta npsA$ strain, demonstrating that the effect is toxin-independent. This data is presented in Reviewer Figure 3.*

Reviewer figure 3: (A) Resulting fecal colonization levels of *E. coli* MDR1 after different time points of colonization. Data represents the mean and SEM one experiments with n=3-5 mice per group. (B) Clearance kinetics of *E. coli* MDR1 after different time points of colonization (clearance = CFU/g below the detection limit in feces).

Minor

- Data on unitigs (Fig4A) seems irrelevant for the rest the main narrative of the manuscript and lacks a better explanation; should not be in a main figure

Author's response: We moved the data on unitigs from Figure 4A to the now revised figure S4A.

- AUC values can be quite subjective and depict completely different growth phenotypes. The growth kinetics should be added as supplementary figures.

Author's response: We agree that AUC values can depict completely different growth phenotypes. Still, they can be used to represent the growth of bacterial strains, especially when we want to compare growth to no or little growth. We decided to show the growth in Figure 4A and S6 in a heatmap representing the AUC, since we would have to show 40 (Figure 4A) and 180 (Figure S6) different plots for the respective carbon sources.

- Several statements regarding the microbiome analysis (including Fig3D, G, H and I) lack proper statistical tests.

Author's response: We revised Figure 3 and included statistical tests for all plots, see comment reviewer 1.

- Throughout the manuscript figure legends need to be improved to explain all relevant aspects represented (e.g. EcN being the yellow dot in Figure 1, what is MHH and why not use MDR1 like in other figures)

Author's Response: Thank you for pointing this out. Due to its origin, we previously referred to the *E. coli* MDR1 strain as the MHH strain but have decided on a clearer nomenclature using MDR1->3. This has been corrected.

- Material and methods need improvement, namely making sure all numbers of animals used for all experiments are reported, also the calculations for cultures used (in particular in the 10:1 ratio, which appears to be inverted between commensal and MDR strains).

Author's response: Animal number used for each experiment are mentioned in the respective figure legends and are visible in the source data file. For the 10:1 ratio 20 μ l of the commensal culture (OD=1) and 10 μ l of the MDR culture (OD=0.2) were added to the germfree cecum content, as described in the material and methods section.

- Present other metrics of diversity, like the Shannon index, for PBS vs MR102 in Figure 3

Author's response: Comparison of α -diversity in observed OTUs and Shannon-index shows significantly higher levels for MR102-treated groups compared to PBS, MR158, MK192 and LK192-treated groups. The data is now included in Figure 3C and D and is mentioned in line 267f.

- Show pre-colonization day for the alpha diversity in Figure 3E

Author's response: We could not include the PBS-treated group for the pre-colonization timepoint for the analysis, since the samples did not have enough reads for sufficient analysis. We decided to show the α -diversity data from day 6, since this is the first timepoint after the antibiotic treatment and the most interesting day regarding the microbiome recovery, as we demonstrate in revised Figure 3B. At the precolonization timepoint (d-1) observed OTUs are <50 in all groups (see reviewer figure 4).

Reviewer figure 4: α -diversity represented by observed OTUs at day -1.

- Show PBS microbiome in pre-colonization time-point (also refer to what day it refers to) in Figure S3

Author's response: *The pre-colonization timepoint refers to day -1. We could not include the PBS microbiome composition, since the samples did not have enough reads for sufficient analysis.*

- 1% cellobiose was supplemented to the mice and not 10% as is depicted in the figure (10g/l)

Author's response: *Thank you. We corrected Figure 4C.*

- Explain if OMM mice were from colonies of mice raised with OMM as microbiota or if these are ex-germ-free mice colonized with OMM communities for the experiments

Author's response: *OMM12 and OMM19 mice were generated by colonization of germfree mice. Germfree mice were gavaged orally and rectally with commercially available bacterial suspension, at a two-day interval. Accordingly, the description in the Material and Methods section was adapted under "Mice".*

- There is no E panel in Supp figure 2, but there a panel E is described in the legend.

Author's response: *This was a mistake, there is no panel E, since this is the Figure 2F.*

- What is the number of mice used for Fig5?

Author's response: *Two independent experiments with n=8-10 mice per group were done. This is now included in the Figure legend and the source data*

Reviewer #4 (Remarks to the Author):

Point to point answer reviewers comments:

Reviewer's response: This information is essential in the manuscript. However, this still leaves the question regarding the selective plating of MDR1 vs competitive strains, i.e. the methods (both for in vitro and in vivo experiments) should specify the combination of antibiotics that were used to exclusively plate MDR1 (supposedly carbapenemase) and the antibiotics that were used to plate for the competitive strain (ampicillin?), including EcN, since ampicillin alone should select both the competitive and the MDR1, correct? Furthermore, was the nonresistance to carbapenemase of the competitive strains tested or just inferred from the genome? The Reviewer Table 1 raises the question of why not test the MR043, since it has a lower FC than both RV228 and MR158. The reason for not including this strain should be specified in the text.

Shouldn't the selected strains with promising results (at the minimum MR102) be tested at least for MDR status, doing an antibiotic profile, since that would be extremely relevant for the potential use as a probiotic?

Author's response: We agree that the information regarding selective plating is essential and have included a sentence in the Methods section (Line 633f); in short, we used ciprofloxacin as an antibiotic. The non-resistance of the commensal isolates was verified through plating. The antibiotic resistance and virulence profile of MR102 were analysed through the genomic analysis, see Table S4/ S5. A more detailed functional analysis of virulence and antibiotic resistance of the described commensal strains, including MR043, is currently ongoing.

Reviewer's response: It is understandable the use of a subset of 10% to select strains as the most protective strains, but it should be avoidable to call all the other strains as non-protective (as seen in table S4), since some protective and non-protective strains would be very similar in their effect on MDR1 (LK039 vs LK001).

Author's response: We renamed the strains in Table S4 to "10% most-protective" and "other 90%" in Table S4. Furthermore, we have included a short explanation in line 152f. Since all genomes and inhibitory factors are listed in the manuscript, the readers will be able to refine the analysis on demand according to their own criteria.

Reviewer's response: There is information in these results that is very relevant to be conveyed to the reader beyond the blind quantification of strains affected by the competition. Namely, the discriminated effect of the combinations compared to the single treatments, such as with EcN and MK01, in which the combination leads to reduction of 3 strains that are unaffected by either EcN or MK01 alone, but on the other hand, 2 strains become unaffected by the combination when MK01 alone was enough to reduce it. Similarly, the combination of MK01 with LK192 results in 3 strains being reduced (that were unaffected by MK01 or LK192 alone), but also results in 2 strains becoming unaffected even though they were reduced by MK01 alone. This loss of protection upon combination with MK01 also happens with MR102 to 2 strains, while 3 are only reduced upon combination. This is relevant because this shows that the treatment (single or in combination) needs to match the MDR strain and not be blindly used.

Author's response: We included two additional sentences in the Results and Discussion section in Line 444f and 561ff, commenting on this observation. We agree that further studies in the future are needed to clarify whether those effects hold true in vivo.

Author's response: [editorial note: reply redacted]

Reviewer's response: Since the experiment has been done and even though it is a "negative" result, it is still a result that should be disclosed, therefore it is advisable that this experiment is added to the manuscript, because it clearly highlights the bigger role of nutrition competition in this model.

Author's response: We agree and included this experiment in Figure S7 and Line 474ff.

Reviewer's note to authors: Please correct the mistake on Fig.3 (panel letters A and B are switched).

Author's response: This has been corrected.